# SPARSITY-CONSTRAINED OPTIMAL TRANSPORT

**Tianlin Liu**[*]
University of Basel

**Joan Puigcerver**
Google Research, Brain team

**Mathieu Blondel**
Google Research, Brain team

## ABSTRACT

Regularized optimal transport (OT) is now increasingly used as a loss or as a matching layer in neural networks. Entropy-regularized OT can be computed using the Sinkhorn algorithm but it leads to fully-dense transportation plans, meaning that all sources are (fractionally) matched with all targets. To address this issue, several works have investigated quadratic regularization instead. This regularization preserves sparsity and leads to unconstrained and smooth (semi) dual objectives, that can be solved with off-the-shelf gradient methods. Unfortunately, quadratic regularization does not give direct control over the cardinality (number of nonzeros) of the transportation plan. We propose in this paper a new approach for OT with explicit cardinality constraints on the transportation plan. Our work is motivated by an application to sparse mixture of experts, where OT can be used to match input tokens such as image patches with expert models such as neural networks. Cardinality constraints ensure that at most $k$ tokens are matched with an expert, which is crucial for computational performance reasons. Despite the non-convexity of cardinality constraints, we show that the corresponding (semi) dual problems are tractable and can be solved with first-order gradient methods. Our method can be thought as a middle ground between unregularized OT (recovered when $k$ is small enough) and quadratically-regularized OT (recovered when $k$ is large enough). The smoothness of the objectives increases as $k$ increases, giving rise to a trade-off between convergence speed and sparsity of the optimal plan.

## 1 INTRODUCTION

Optimal transport (OT) distances (a.k.a. Wasserstein or earth mover's distances) are a powerful computational tool to compare probability distributions and have found widespread use in machine learning (Solomon et al., 2014; Kusner et al., 2015; Arjovsky et al., 2017). While OT distances exhibit a unique ability to capture the geometry of the data, their applicability has been largely hampered by their high computational cost. Indeed, computing OT distances involves a linear program, which takes super-cubic time to solve using state-of-the-art network-flow algorithms (Kennington & Helgason, 1980; Ahuja et al., 1988). In addition, these algorithms are challenging to implement and are not GPU or TPU friendly. An alternative approach consists instead in solving the so-called semi-dual using (stochastic) subgradient methods (Carlier et al., 2015) or quasi-Newton methods (Mérigot, 2011; Kitagawa et al., 2019). However, the semi-dual is a nonsmooth, piecewise-linear function, which can lead to slow convergence in practice.

For all these reasons, the machine learning community has now largely switched to regularized OT. Popularized by Cuturi (2013), entropy-regularized OT can be computed using the Sinkhorn algorithm (1967) and is differentiable w.r.t. its inputs, enabling OT as a differentiable loss (Cuturi, 2013; Feydy et al., 2019) or as a layer in a neural network (Genevay et al., 2019; Sarlin et al., 2020; Sander et al., 2022). A disadvantage of entropic regularization, however, is that it leads to fully-dense transportation plans. This is problematic in applications where it is undesirable to (fractionally) match all sources with all targets, e.g., for interpretability or for computational cost reasons. To address this issue, several works have investigated quadratic regularization instead (Dessein et al., 2018; Blondel et al., 2018; Lorenz et al., 2021). This regularization preserves sparsity and leads to unconstrained and smooth (semi) dual objectives, solvable with off-the-shelf algorithms. Unfortunately, it does not give direct control over the cardinality (number of nonzeros) of the transportation plan.

---

[*]Work done during an internship at Google Research, Brain Team.

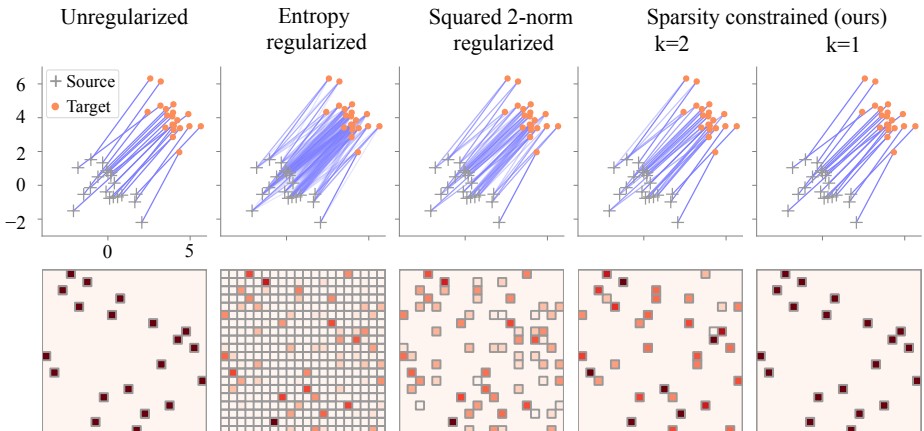

Figure 1: **OT formulation comparison** ($m = n = 20$ points), with squared Euclidean distance cost, and with uniform source and target distributions. The unregularized OT plan is maximally sparse and contains at most $m+n-1$ nonzero elements. On the contrary, with entropy-regularized OT, plans are always fully dense, meaning that all points are fractionally matched with one another (nonzeros of a transportation plan are indicated by small squares). Squared 2-norm (quadratically) regularized OT preserves sparsity but the number of nonzero elements cannot be directly controlled. Our proposed sparsity-constrained OT allows us to set a maximum number of nonzeros $k$ per column. It recovers unregularized OT in the limit case $k = 1$ (Proposition 4) and quadratically-regularized OT when $k$ is large enough. It can be computed using solvers such as LBFGS or ADAM.

In this paper, we propose a new approach for OT with explicit cardinality constraints on the columns of the transportation plan. Our work is motivated by an application to sparse mixtures of experts, in which we want each token (e.g. a word or an image patch) to be matched with at most $k$ experts (e.g., multilayer perceptrons). This is critical for computational performance reasons, since the cost of processing a token is proportional to the number of experts that have been selected for it. Despite the nonconvexity of cardinality constraints, we show that the corresponding dual and semi-dual problems are tractable and can be solved with first-order gradient methods. Our method can be thought as a middle ground between unregularized OT (recovered when $k$ is small enough) and quadratically-regularized OT (recovered when $k$ is large enough). We empirically show that the dual and semi-dual are increasingly smooth as $k$ increases, giving rise to a trade-off between convergence speed and sparsity. The rest of the paper is organized as follows.

- We review related work in §2 and existing work on OT with convex regularization in §3.

- We propose in §4 a framework for OT with nonconvex regularization, based on the dual and semi-dual formulations. We study the weak duality and the primal interpretation of these formulations.

- We apply our framework in §5 to OT with cardinality constraints. We show that the dual and semi-dual formulations are tractable and that smoothness of the objective increases as $k$ increases. We show that our approach is equivalent to using squared $k$-support norm regularization in the primal.

- We validate our framework in §6 and in Appendix A through a variety of experiments.

**Notation and convex analysis tools.** Given a matrix $T \in \mathbb{R}^{m \times n}$, we denote its columns by $\boldsymbol{t}_j \in \mathbb{R}^m$ for $j \in [n]$. We denote the non-negative orthant by $\mathbb{R}^m_+$ and the non-positive orthant by $\mathbb{R}^m_-$. We denote the probability simplex by $\triangle^m := \{\boldsymbol{p} \in \mathbb{R}^m_+ : \langle \boldsymbol{p}, \mathbf{1} \rangle = 1\}$. We will also use $b\triangle^m$ to denote the set $\{\boldsymbol{t} \in \mathbb{R}^m_+ : \langle \boldsymbol{t}, \mathbf{1} \rangle = b\}$. The convex conjugate of a function $f \colon \mathbb{R}^m \to \mathbb{R} \cup \{\infty\}$ is defined by $f^*(\boldsymbol{s}) := \sup_{\boldsymbol{t} \in \text{dom}(f)} \langle \boldsymbol{s}, \boldsymbol{t} \rangle - f(\boldsymbol{t})$. It is well-known that $f^*$ is convex (even if $f$ is not). If the solution is unique, then its gradient is $\nabla f^*(\boldsymbol{s}) = \text{argmax}_{\boldsymbol{t} \in \text{dom}(f)} \langle \boldsymbol{s}, \boldsymbol{t} \rangle - f(\boldsymbol{t})$. If the solution is not unique, then we obtain a subgradient. We denote the indicator function of a set $\mathcal{C}$ by $\delta_{\mathcal{C}}$, i.e., $\delta_{\mathcal{C}}(\boldsymbol{t}) = 0$ if $\boldsymbol{t} \in \mathcal{C}$ and $\delta_{\mathcal{C}}(\boldsymbol{t}) = \infty$ otherwise. We denote the Euclidean projection onto the set $\mathcal{C}$ by $\text{proj}_{\mathcal{C}}(\boldsymbol{s}) = \text{argmin}_{\boldsymbol{t} \in \mathcal{C}} \|\boldsymbol{s} - \boldsymbol{t}\|_2^2$. The projection is unique when $\mathcal{C}$ is convex, while it may not be when $\mathcal{C}$ is nonconvex. We use $[\cdot]_+$ to denote the non-negative part, evaluated element-wise. Given a vector $\boldsymbol{s} \in \mathbb{R}^m$, we use $s_{[i]}$ to denote its $i$-th largest value, i.e., $s_{[1]} \geq \cdots \geq s_{[m]}$.

## 2 RELATED WORK

**Sparse optimal transport.** OT with arbitrary strongly convex regularization is studied by Dessein et al. (2018) and Blondel et al. (2018). More specifically, quadratic regularization was studied in the discrete (Blondel et al., 2018; Roberts et al., 2017) and continuous settings (Lorenz et al., 2021). Although it is known that quadratic regularization leads to sparse transportation plans, it does not enable explicit control of the cardinality (maximum number of nonzero elements), as we do. In this work, we study the nonconvex regularization case and apply it to cardinality-constrained OT.

**Sparse projections.** In this paper, we use $k$-sparse projections as a core building block of our framework. Sparse projections on the simplex and on the non-negative orthant were studied by Kyrillidis et al. (2013) and Bolte et al. (2014), respectively. These studies were later extended to more general sets (Beck & Hallak, 2016). On the application side, sparse projections on the simplex were used for structured prediction (Pillutla et al., 2018; Blondel et al., 2020), for marginalizing over discrete variables (Correia et al., 2020) and for Wasserstein $K$-means (Fukunaga & Kasai, 2021).

**Sparse mixture of experts (MoE).** In contrast to usual deep learning models where all parameters interact with all inputs, a sparse MoE model activates only a small part of the model ("experts") in an input-dependent manner, thus reducing the overall computational cost of the model. Sparse MoEs have been tremendously successful in scaling up deep learning architectures in tasks including computer vision (Riquelme et al., 2021), natural language processing (Shazeer et al., 2017; Lewis et al., 2021; Lepikhin et al., 2021; Roller et al., 2021; Fedus et al., 2022b; Clark et al., 2022), speech processing (You et al., 2022), and multimodal learning (Mustafa et al., 2022). In addition to reducing computational cost, sparse MoEs have also shown other benefits, such as an enhancement in adversarial robustness (Puigcerver et al., 2022). See Fedus et al. (2022a) for a recent survey. Crucial to a sparse MoE model is its **routing mechanism** that decides which experts get which inputs. Transformer-based MoE models typically route individual tokens (embedded words or image patches). To balance the assignments of tokens to experts, recent works cast the assignment problem as entropy-regularized OT (Kool et al., 2021; Clark et al., 2022). We go beyond entropy-regularized OT and show that sparsity-constrained OT yields a more natural and effective router.

## 3 OPTIMAL TRANSPORT WITH CONVEX REGULARIZATION

In this section, we review OT with convex regularization, which also includes the unregularized case. For a comprehensive survey on computational OT, see (Peyré & Cuturi, 2019).

**Primal formulation.** We focus throughout this paper on OT between discrete probability distributions $\boldsymbol{a} \in \triangle^m$ and $\boldsymbol{b} \in \triangle^n$. Rather than performing a pointwise comparison of the distributions, OT distances compute the minimal effort, according to some ground cost, for moving the probability mass of one distribution to the other. Recent applications of OT in machine learning typically add regularization on the transportation plan $T$. In this section, we apply **convex** regularization $\Omega \colon \mathbb{R}_+^m \to \mathbb{R}_+ \cup \{\infty\}$ **separately** on the columns $\boldsymbol{t}_j \in \mathbb{R}_+^m$ of $T$ and consider the primal formulation

$$P_\Omega(\boldsymbol{a}, \boldsymbol{b}, C) \coloneqq \min_{T \in \mathcal{U}(\boldsymbol{a}, \boldsymbol{b})} \langle T, C \rangle + \sum_{j=1}^n \Omega(\boldsymbol{t}_j), \tag{1}$$

where $C \in \mathbb{R}_+^{m \times n}$ is a cost matrix and $\mathcal{U}(\boldsymbol{a}, \boldsymbol{b}) \coloneqq \{T \in \mathbb{R}_+^{m \times n} \colon T\mathbf{1}_n = \boldsymbol{a}, T^\top \mathbf{1}_m = \boldsymbol{b}\}$ is the transportation polytope, which can be interpreted as the set of all joint probability distributions with marginals $\boldsymbol{a}$ and $\boldsymbol{b}$. It includes the Birkhoff polytope as a special case when $m = n$ and $\boldsymbol{a} = \boldsymbol{b} = \frac{\mathbf{1}_m}{m}$.

**Dual and semi-dual formulations.** Let us denote

$$\Omega_+^*(\boldsymbol{s}) \coloneqq (\Omega + \delta_{\mathbb{R}_+^m})^*(\boldsymbol{s}) = \max_{\boldsymbol{t} \in \mathbb{R}_+^m} \langle \boldsymbol{s}, \boldsymbol{t} \rangle - \Omega(\boldsymbol{t}) \tag{2}$$

and

$$\Omega_b^*(\boldsymbol{s}) \coloneqq (\Omega + \delta_{b\triangle^m})^*(\boldsymbol{s}) = \max_{\boldsymbol{t} \in b\triangle^m} \langle \boldsymbol{s}, \boldsymbol{t} \rangle - \Omega(\boldsymbol{t}). \tag{3}$$

| | $\Omega(\boldsymbol{t})$ | $\Omega^*_+(\boldsymbol{s})$ | $\Omega^*_b(\boldsymbol{s})$ |
|---|---|---|---|
| Unregularized | $0$ | $\delta_{\mathbb{R}^m_-}(\boldsymbol{s})$ | $b \max_{i \in [m]} s_i$ |
| Negentropy | $\langle \boldsymbol{t}, \log \boldsymbol{t} \rangle$ | $\sum_{i=1}^m e^{s_i - 1}$ | $\log \sum_{i=1}^m e^{s_i} - b$ |
| Squared 2-norm | $\frac{1}{2}\|\boldsymbol{t}\|_2^2$ | $\frac{1}{2}\sum_{i=1}^m [s_i]_+^2$ | $\frac{1}{2}\sum_{i=1}^m \mathbb{1}_{s_i \geq \theta}(s_i^2 - \theta^2)$ |
| Sparsity-constrained (top-$k$) | $\frac{1}{2}\|\boldsymbol{t}\|_2^2 + \delta_{\mathcal{B}_k}(\boldsymbol{t})$ | $\frac{1}{2}\sum_{i=1}^k [s_{[i]}]_+^2$ | $\frac{1}{2}\sum_{i=1}^k \mathbb{1}_{s_{[i]} \geq \tau}(s_{[i]}^2 - \tau^2)$ |
| Sparsity-constrained (top-1) | $\frac{1}{2}\|\boldsymbol{t}\|_2^2 + \delta_{\mathcal{B}_1}(\boldsymbol{t})$ | $\frac{1}{2}\max_{i \in [m]}[s_i]_+^2$ | $b \max_{i \in [m]} s_i - \frac{\gamma}{2}b^2$ |

Table 1: Summary of the conjugate expressions (2) and (3) for various choices of $\Omega$. Here, $\theta$ and $\tau$ are such that $\sum_{i=1}^m [s_i - \theta]_+$ and $\sum_{i=1}^k [s_{[i]} - \tau]_+$ sum to $b$ (§5), where $s_{[i]}$ denotes the $i$-th largest entry of the vector $\boldsymbol{s} \in \mathbb{R}^m$. The top-$k$ and top-1 expressions above assume no ties in $\boldsymbol{s}$.

The dual and semi-dual corresponding to (1) can then be written (Blondel et al., 2018) as

$$D_\Omega(\boldsymbol{a}, \boldsymbol{b}, C) := \max_{\boldsymbol{\alpha} \in \mathbb{R}^m, \boldsymbol{\beta} \in \mathbb{R}^n} \langle \boldsymbol{\alpha}, \boldsymbol{a} \rangle + \langle \boldsymbol{\beta}, \boldsymbol{b} \rangle - \sum_{j=1}^n \Omega^*_+(\boldsymbol{\alpha} + \beta_j \mathbf{1}_m - \boldsymbol{c}_j) \tag{4}$$

and

$$S_\Omega(\boldsymbol{a}, \boldsymbol{b}, C) := \max_{\boldsymbol{\alpha} \in \mathbb{R}^m} \langle \boldsymbol{\alpha}, \boldsymbol{a} \rangle - P^*_\Omega(\boldsymbol{\alpha}, \boldsymbol{b}, C) = \max_{\boldsymbol{\alpha} \in \mathbb{R}^m} \langle \boldsymbol{\alpha}, \boldsymbol{a} \rangle - \sum_{j=1}^n \Omega^*_{b_j}(\boldsymbol{\alpha} - \boldsymbol{c}_j), \tag{5}$$

where $P^*_\Omega$ denotes the conjugate in the first argument. When $\Omega$ is convex (which also includes the unregularized case $\Omega = 0$), by strong duality, we have that $P_\Omega(\boldsymbol{a}, \boldsymbol{b}, C) = D_\Omega(\boldsymbol{a}, \boldsymbol{b}, C) = S_\Omega(\boldsymbol{a}, \boldsymbol{b}, C)$ for all $\boldsymbol{a} \in \triangle^m$, $\boldsymbol{b} \in \triangle^n$ and $C \in \mathbb{R}^{m \times n}_+$.

**Computation.** With $\Omega = 0$ (without regularization), then (2) becomes the indicator function of the non-positive orthant, leading to the constraint $\alpha_i + \beta_j \leq c_{i,j}$. The dual (4) is then a constrained linear program and the most commonly used algorithm is the network flow solver. On the other hand, (3) becomes a max operator, leading to the so-called $c$-transform $\beta_j = \min_{i \in [m]} c_{i,j} - \alpha_i$ for all $j \in [n]$. The semi-dual (5) is then unconstrained, but it is a nonsmooth piecewise linear function.

The key advantage of introducing strongly convex regularization is that it makes the corresponding (semi) dual easier to solve. Indeed, (2) and (3) become "soft" constraints and max operators.

In particular, when $\Omega$ is Shannon's negentropy $\Omega(\boldsymbol{t}) = \gamma \langle \boldsymbol{t}, \log \boldsymbol{t} \rangle$, where $\gamma$ controls the regularization strength, then (2) and (3) rely on the exponential and log-sum-exp operations. It is well known that the primal (1) can then be solved using Sinkhorn's algorithm (Cuturi, 2013), which amounts to using a block coordinate ascent scheme w.r.t. $\boldsymbol{\alpha} \in \mathbb{R}^m$ and $\boldsymbol{\beta} \in \mathbb{R}^n$ in the dual (4). As pointed out in Blondel et al. (2018), the semi-dual is smooth (i.e., with Lipschitz gradients) but the dual is not.

When $\Omega$ is the quadratic regularization $\Omega(\boldsymbol{t}) = \frac{\gamma}{2}\|\boldsymbol{t}\|_2^2$, then as shown in Blondel et al. (2018, Table 1), (2) and (3) rely on the squared relu and on the projection onto the simplex. However, it is shown empirically that a block coordinate ascent scheme in the dual (4) converges slowly. Instead, Blondel et al. (2018) propose to use LBFGS both to solve the dual and the semi-dual. Both the dual and the semi-dual are smooth (Blondel et al., 2018), i.e., with Lipschitz gradients. For both types of regularization, when $\gamma \to 0$, we recover unregularized OT.

**Recovering a plan.** If $\Omega$ is strictly convex, the unique optimal solution $T^\star$ of (1) can be recovered from an optimal solution $(\boldsymbol{\alpha}^\star, \boldsymbol{\beta}^\star)$ of the dual (4) by

$$\boldsymbol{t}_j^\star = \nabla \Omega^*_+(\boldsymbol{\alpha}^\star + \beta_j^\star \mathbf{1}_m - \boldsymbol{c}_j) \quad \forall j \in [n] \tag{6}$$

or from an optimal solution $\boldsymbol{\alpha}^\star$ of the semi-dual (5) by

$$\boldsymbol{t}_j^\star = \nabla \Omega^*_{b_j}(\boldsymbol{\alpha}^\star - \boldsymbol{c}_j) \quad \forall j \in [n]. \tag{7}$$

If $\Omega$ is convex but not strictly so, recovering $T^\star$ is more involved; see Appendix B.2 for details.

## 4 OPTIMAL TRANSPORT WITH NONCONVEX REGULARIZATION

In this section, we again focus on the primal formulation (1), but now study the case when the regularization $\Omega\colon \mathbb{R}_+^m \to \mathbb{R}_+ \cup \{\infty\}$ is **nonconvex**.

**Concavity.** It is well-known that the conjugate function is always convex, even when the original function is not. As a result, even if the conjugate expressions (2) and (3) involve nonconcave maximization problems in the variable $t$, they induce convex functions in the variable $s$. We can therefore make the following elementary remark: the dual (4) and the semi-dual (5) are **concave** maximization problems, **even if** $\Omega$ is nonconvex. This means that we can solve them to arbitrary precision **as long as** we know how to compute the conjugate expressions (2) and (3). This is generally hard but we will see in §5 a setting in which these expressions can be computed **exactly**. We remark that the identity $S_\Omega(a, b, C) = \max_{\alpha \in \mathbb{R}^m} \langle \alpha, a \rangle - P_\Omega^*(\alpha, b, C)$ still holds even when $\Omega$ is nonconvex.

**The semi-dual upper-bounds the dual.** Of course, if $\Omega$ is nonconvex, only weak duality holds, i.e., the dual (4) and semi-dual (5) are lower bounds of the primal (1). The next proposition clarifies that the semi-dual is actually an upper-bound for the dual (a proof is given in Appendix B.1).

---

**Proposition 1.** *Weak duality*

*Let $\Omega\colon \mathbb{R}_+^m \to \mathbb{R}_+ \cup \{\infty\}$ (potentially nonconvex). For all $a \in \triangle^m$, $b \in \triangle^n$ and $C \in \mathbb{R}_{m \times n}^+$*

$$D_\Omega(a, b, C) \le S_\Omega(a, b, C) \le P_\Omega(a, b, C).$$

---

Therefore, if the goal is to compute approximately $P_\Omega(a, b, C)$, which involves an intractable nonconvex problem in general, it may be advantageous to use $S_\Omega(a, b, C)$ as a proxy, rather than $D_\Omega(a, b, C)$. However, for the specific choice of $\Omega$ in §5, we will see that $D_\Omega(a, b, C)$ and $S_\Omega(a, b, C)$ actually coincide, i.e., $D_\Omega(a, b, C) = S_\Omega(a, b, C) \le P_\Omega(a, b, C)$.

**Recovering a plan.** Many times, the goal is not to compute the quantity $P_\Omega(a, b, C)$ itself, but rather the associated OT plan. If $\Omega$ is nonconvex, this is again intractable due to the nonconvex nature of the problem. As an approximation, given an optimal solution $(\alpha^\star, \beta^\star)$ of the dual or an optimal solution $\alpha^\star$ of the semi-dual, we propose to recover a transportation plan with (6) and (7), just like we would do in the convex $\Omega$ case. The following proposition clarifies that the optimal transportation plan $T^\star$ that we get corresponds to a convex relaxation of the original nonconvex problem (1). A proof is given in Appendix B.3.

---

**Proposition 2.** *Primal interpretation*

*Let $\Omega\colon \mathbb{R}_+^m \to \mathbb{R}_+ \cup \{\infty\}$ (potentially nonconvex). For all $a \in \triangle^m$, $b \in \triangle^n$ and $C \in \mathbb{R}_+^{m \times n}$*

$$D_\Omega(a, b, C) = \min_{\substack{T \in \mathbb{R}^{m \times n} \\ T\mathbf{1}_n = a \\ T^\top \mathbf{1}_m = b}} \langle T, C \rangle + \sum_{j=1}^n \Omega_+^{**}(t_j) = P_{\Omega^{**}}(a, b, C)$$

$$S_\Omega(a, b, C) = \min_{\substack{T \in \mathbb{R}^{m \times n} \\ T\mathbf{1}_n = a}} \langle T, C \rangle + \sum_{j=1}^n \Omega_{b_j}^{**}(t_j) = P_{\Omega^{**}}(a, b, C).$$

---

In the above, $f^{**}$ denotes the biconjugate of $f$, the tightest convex lower bound of $f$. When $\Omega$ is nonconvex, deriving an expression for $\Omega_+^{**}$ and $\Omega_{b_j}^{**}$ could be challenging in general. Fortunately, for the choice of $\Omega$ in §5, we are able to do so. When a function is convex and closed, its biconjugate is itself. As a result, if $\Omega$ is a convex and closed function, we recover $P_\Omega(a, b, C) = D_\Omega(a, b, C) = S_\Omega(a, b, C)$ for all $a \in \triangle^m$, $b \in \triangle^n$ and $C \in \mathbb{R}_+^{m \times n}$.

**Summary: proposed method.** To approximately solve the primal OT objective (1) when $\Omega$ is nonconvex, we proposed to solve the dual (4) or the semi-dual (5), which by Proposition 1 lower-bound the primal. We do so by solving the concave maximization problems in (4) and (5) by gradient-based algorithms, such as LBFGS (Liu & Nocedal, 1989) or ADAM (Kingma & Ba, 2015). When a transportation plan is needed, we recover it from (6) and (7), as we would do with convex $\Omega$.

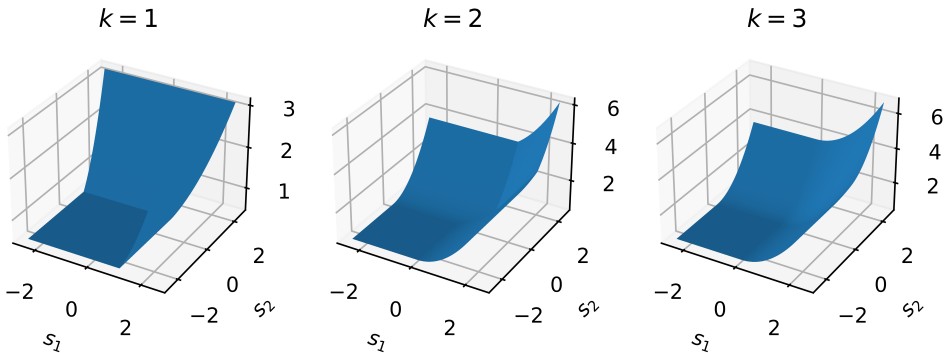

Figure 2: **Increasing $k$ increases smoothness.** Let $\boldsymbol{s} = (s_1, s_2, 0, 1)$. We visualize $\Omega_+^*(\boldsymbol{s})$, defined in (10) and derived in Table 1, when varying $s_1$ and $s_2$. It can be interpreted as a relaxation of the indicator function of the non-positive orthant. The conjugate $\Omega_b^*(\boldsymbol{s})$ (not shown) can be interpreted as a relaxed max operator, scaled by $b$. In both cases, the smoothness increases when $k$ increases.

Proposition 2 clarifies what objective function this plan optimally solves. When learning with OT as a loss, it is necessary to differentiate through $S_\Omega(\boldsymbol{a}, \boldsymbol{b}, C)$. From Danskin's theorem, the gradients $\nabla_{\boldsymbol{a}} S_\Omega(\boldsymbol{a}, \boldsymbol{b}, C) \in \mathbb{R}^m$ and $\nabla_C S_\Omega(\boldsymbol{a}, \boldsymbol{b}, C) \in \mathbb{R}^{m \times n}$ are given by $\boldsymbol{\alpha}^\star$ and $T^\star$ from (7), respectively.

## 5 QUADRATICALLY-REGULARIZED OT WITH SPARSITY CONSTRAINTS

In this section, we build upon §4 to develop a regularized OT formulation with sparsity constraints.

**Formulation.** Formally, given $\boldsymbol{t} \in \mathbb{R}^m$, let us define the $\ell_0$ pseudo norm by

$$\|\boldsymbol{t}\|_0 := |\{t_j \neq 0 : j \in [m]\}|,$$

i.e., the number of nonzero elements in $\boldsymbol{t}$. For $k \in \{1, \ldots, m\}$, we denote the $\ell_0$ level sets by

$$\mathcal{B}_k := \{\boldsymbol{t} \in \mathbb{R}^m : \|\boldsymbol{t}\|_0 \leq k\}.$$

Our goal in this section is then to approximately solve the following quadratically-regularized optimal transport problem with cardinality constraints on the columns of $T$

$$\min_{\substack{T \in \mathcal{U}(\boldsymbol{a}, \boldsymbol{b}) \\ T \in \mathcal{B}_k \times \cdots \times \mathcal{B}_k}} \langle T, C \rangle + \frac{\gamma}{2} \|T\|_2^2, \tag{8}$$

where $\gamma > 0$ controls the regularization strength and where $k$ is assumed large enough to make the problem feasible. Problem (8) is a special case of (1) with the nonconvex regularization

$$\Omega = \frac{\gamma}{2} \| \cdot \|_2^2 + \delta_{\mathcal{B}_k}. \tag{9}$$

We can therefore apply the methodology outlined in §4. If the cardinality constraints need to be applied to the rows instead of to the columns, we simply transpose the problem.

**Computation.** We recall that in order to solve the dual (4) or the semi-dual (5), the main quantities that we need to be able to compute are the conjugate expressions (2) and (3), as well as their gradients. While this is intractable for general nonconvex $\Omega$, with the choice of $\Omega$ in (9), we obtain

$$\Omega_+^*(\boldsymbol{s}) = \max_{\boldsymbol{t} \in \mathbb{R}_+^m \cap \mathcal{B}_k} \langle \boldsymbol{s}, \boldsymbol{t} \rangle - \frac{1}{2} \|\boldsymbol{t}\|_2^2 \tag{10}$$

$$\Omega_b^*(\boldsymbol{s}) = \max_{\boldsymbol{t} \in b \triangle^m \cap \mathcal{B}_k} \langle \boldsymbol{s}, \boldsymbol{t} \rangle - \frac{1}{2} \|\boldsymbol{t}\|_2^2,$$

where, without loss of generality, we assumed $\gamma = 1$. Indeed, when $\gamma \neq 1$, we can simply use the property $(\gamma f)^* = \gamma f^*(\cdot/\gamma)$. From the envelope theorem of Rockafellar & Wets (2009, Theorem

10.31), the gradients are given by the corresponding argmax problems and we obtain

$$\nabla \Omega_+^*(\boldsymbol{s}) = \underset{\boldsymbol{t} \in \mathbb{R}_+^m \cap \mathcal{B}_k}{\operatorname{argmax}} \langle \boldsymbol{s}, \boldsymbol{t} \rangle - \frac{1}{2}\|\boldsymbol{t}\|_2^2 = \operatorname{proj}_{\mathbb{R}_+^m \cap \mathcal{B}_k}(\boldsymbol{s})$$

$$\nabla \Omega_b^*(\boldsymbol{s}) = \underset{\boldsymbol{t} \in b\triangle^m \cap \mathcal{B}_k}{\operatorname{argmax}} \langle \boldsymbol{s}, \boldsymbol{t} \rangle - \frac{1}{2}\|\boldsymbol{t}\|_2^2 = \operatorname{proj}_{b\triangle^m \cap \mathcal{B}_k}(\boldsymbol{s}).$$

Therefore, computing an optimal solution $\boldsymbol{t}^\star$ reduces to the $k$-sparse projections of $\boldsymbol{s}$ onto the non-negative orthant and onto the simplex (scaled by $b > 0$), respectively. When $\boldsymbol{t}^\star$ is not unique (i.e., $\boldsymbol{s}$ contains ties), the argmax is set-valued. We discuss this situation in more details in Appendix B.2.

Fortunately, despite the nonconvexity of the set $\mathcal{B}_k$, it turns out that both $k$-sparse projections can be computed **exactly** (Kyrillidis et al., 2013; Bolte et al., 2014; Beck & Hallak, 2016) by composing the unconstrained projection onto the original set with a top-$k$ operation:

$$\operatorname{proj}_{\mathbb{R}_+^m \cap \mathcal{B}_k}(\boldsymbol{s}) = \operatorname{proj}_{\mathbb{R}_+^m}(\operatorname{topk}(\boldsymbol{s})) = [\operatorname{topk}(\boldsymbol{s})]_+ \tag{11}$$

$$\operatorname{proj}_{b\triangle^m \cap \mathcal{B}_k}(\boldsymbol{s}) = \operatorname{proj}_{b\triangle^m}(\operatorname{topk}(\boldsymbol{s})) = [\operatorname{topk}(\boldsymbol{s}) - \tau \mathbf{1}_m]_+, \tag{12}$$

for some normalization constant $\tau \in \mathbb{R}$, such that the solution sums to $b$. Here, $\operatorname{topk}(\boldsymbol{s})$ is defined such that $[\operatorname{topk}(\boldsymbol{s})]_i = s_i$ if $s_i$ is in the top-$k$ elements of $\boldsymbol{s}$ and $[\operatorname{topk}(\boldsymbol{s})]_i = -\infty$ otherwise. The $k$-sparse projection on the simplex is also known as top-$k$ sparsemax (Pillutla et al., 2018; Blondel et al., 2020; Correia et al., 2020). Plugging these solutions back into $\langle \boldsymbol{s}, \boldsymbol{t} \rangle - \frac{1}{2}\|\boldsymbol{t}\|_2^2$, we obtain the expressions in Table 1 (a proof is given in Appendix B.4).

Computing (11) or (12) requires a top-$k$ sort and the projection of a vector of size at most $k$. A top-$k$ sort can be computed in $O(m \log k)$ time, $\operatorname{proj}_{\mathbb{R}_+^m}$ simply amounts to the non-negative part $[\cdot]_+$ and computing $\tau$, as needed for $\operatorname{proj}_{b\triangle^m}$, can be computed in $O(k)$ time (Michelot, 1986; Duchi et al., 2008), by reusing the top-$k$ sort. We have thus obtained an efficient way to compute the conjugates (2) and (3). The total complexity per LBFGS or ADAM iteration is $O(mn \log k)$.

**Recovering a plan.** Assuming no ties in $\boldsymbol{\alpha}^\star + \beta_j^\star \mathbf{1}_m - \boldsymbol{c}_j$ or in $\boldsymbol{\alpha}^\star - \boldsymbol{c}_j$, the corresponding column of the transportation plan is uniquely determined by $\nabla \Omega_+^*(\boldsymbol{\alpha}^\star + \beta_j^\star \mathbf{1}_m - \boldsymbol{c}_j)$ or $\nabla \Omega_{b_j}^*(\boldsymbol{\alpha}^\star - \boldsymbol{c}_j)$, respectively. From (11) and (12), this column belongs to $\mathcal{B}_k$. In case of ties, ensuring that the plan belongs to $\mathcal{U}(\boldsymbol{a}, \boldsymbol{b})$ requires to solve a system of linear equations, as detailed in Appendix B.2. Unfortunately, the columns may fail to belong to $\mathcal{B}_k$ in this situation.

**Biconjugates and primal interpretation.** As we discussed in §4 and Proposition 2, the biconjugates $\Omega_+^{**}$ and $\Omega_b^{**}$ allow us to formally define what primal objective the transportation plans obtained by (6) and (7) optimally solve when $\Omega$ is nonconvex. Fortunately, for the case of $\Omega$ defined in (9), we are able to derive an actual expression. Let us define the squared $k$-support norm by

$$\Omega^{**}(\boldsymbol{t}) = \Psi(\boldsymbol{t}) := \frac{1}{2} \min_{\boldsymbol{\lambda} \in \mathbb{R}^m} \sum_{i=1}^m \frac{t_i^2}{\lambda_i} \quad \text{s.t.} \quad \langle \boldsymbol{\lambda}, \mathbf{1} \rangle = k, 0 < \lambda_i \le 1 \quad \forall i \in [m]. \tag{13}$$

The $k$-support norm is known to be the tightest convex relaxation of the $\ell_0$ pseudo norm over the $\ell_2$ unit ball and can be computed in $O(m \log m)$ time (Argyriou et al., 2012; McDonald et al., 2016). We then have the following proposition, proved in Appendix B.6.

---

**Proposition 3.** *Biconjugate and primal interpretation*

*With $\Omega$ defined in (9), we have*

$$\Omega_+^{**}(\boldsymbol{t}) = \Psi(\boldsymbol{t}) \text{ if } \boldsymbol{t} \in \mathbb{R}_+^m, \ \Omega_+^{**}(\boldsymbol{t}) = \infty \text{ otherwise.}$$
$$\Omega_b^{**}(\boldsymbol{t}) = \Psi(\boldsymbol{t}) \text{ if } \boldsymbol{t} \in b\triangle^m, \ \Omega_b^{**}(\boldsymbol{t}) = \infty \text{ otherwise.}$$

*Therefore, with $\Omega$ defined in (9), we have for all $\boldsymbol{a} \in \triangle^m$, $\boldsymbol{b} \in \triangle^n$ and $C \in \mathbb{R}_+^{m \times n}$*

$$D_\Omega(\boldsymbol{a}, \boldsymbol{b}, C) = S_\Omega(\boldsymbol{a}, \boldsymbol{b}, C) = P_\Psi(\boldsymbol{a}, \boldsymbol{b}, C) \le P_\Omega(\boldsymbol{a}, \boldsymbol{b}, C).$$

*The last inequality is an equality if there are no ties in $\boldsymbol{\alpha}^\star + \beta_j^\star \mathbf{1}_m - \boldsymbol{c}_j$ or in $\boldsymbol{\alpha}^\star - \boldsymbol{c}_j \ \forall j \in [n]$.*

---

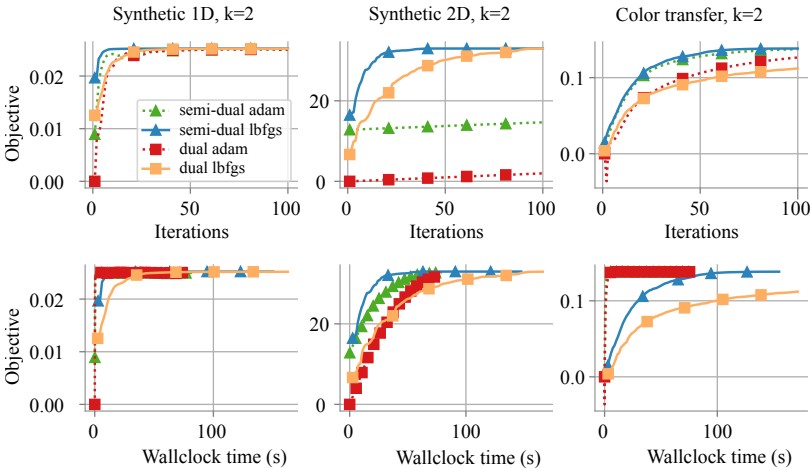

Figure 3: Solver comparison for the semi-dual and dual formulation of sparsity-constrained OT with $k = 2$ on different datasets. Columns correspond to datasets used in Figure 1, Figure 5, and Figure 7.

In other words, our dual and semi-dual approaches based on the nonconvex $\Omega$ are equivalent to using the convex relaxation $\Psi$ as regularization in the primal! We believe that the biconjugate expressions in Proposition 3 are of independent interest and could be useful in other works. For instance, it shows that top-$k$ sparsemax can be alternatively viewed as an argmax regularized with $\Psi$.

**Limit cases and smoothness.** Let $T^\star$ be the solution of the quadratically-regularized OT (without cardinality constraints). If $k \geq \|\boldsymbol{t}_j^\star\|_0$ for all $j \in [n]$, then the constraint $\|\boldsymbol{t}_j\|_0 \leq k$ in (8) is vacuous, and therefore our formulation recovers the quadratically-regularized one. Since $\Omega$ is strongly convex in this case, both conjugates $\Omega_+^*$ and $\Omega_b^*$ are smooth (i.e., with Lipschitz gradients), thanks to the duality between strong convexity and smoothness (Hiriart-Urruty & Lemaréchal, 1993). On the other extreme, when $k = 1$, we have the following (a proof is given in Appendix B.7).

---

**Proposition 4.** *Limit cases*

*With $\Omega$ defined in* (9) *and $k = 1$, we have for all $\boldsymbol{s} \in \mathbb{R}^m$*

$$\Omega_+^*(\boldsymbol{s}) = \frac{1}{2\gamma} \max_{i \in [m]} [s_i]_+^2 \quad and \quad \Omega_b^*(\boldsymbol{s}) = b \max_{i \in [m]} s_i - \frac{\gamma}{2} b^2.$$

*We then have for all $\boldsymbol{a} \in \triangle^m$, $\boldsymbol{b} \in \triangle^n$ and $C \in \mathbb{R}_+^{m \times n}$,*

$$D_\Omega(\boldsymbol{a}, \boldsymbol{b}, C) = S_\Omega(\boldsymbol{a}, \boldsymbol{b}, C) = S_0(\boldsymbol{a}, \boldsymbol{b}, C) + \frac{\gamma}{2} \|\boldsymbol{b}\|_2^2 = P_0(\boldsymbol{a}, \boldsymbol{b}, C) + \frac{\gamma}{2} \|\boldsymbol{b}\|_2^2.$$

---

When $m < n$, it is infeasible to satisfy both the marginal and the 1-sparsity constraints. Proposition 4 shows that our (semi) dual formulations reduce to unregularized OT in this "degenerate" case. As illustrated in Figure 2, the conjugates $\Omega_+^*$ and $\Omega_b^*$ become increasingly smooth as $k$ increases. We therefore interpolate between unregularized OT ($k$ small enough) and quadratically-regularized OT ($k$ large enough), with the dual and semi-dual being increasingly smooth as $k$ increases.

## 6   EXPERIMENTS

### 6.1   SOLVER AND OBJECTIVE COMPARISON

We compared two solvers, LBFGS (Liu & Nocedal, 1989) and ADAM (Kingma & Ba, 2015) for maximizing the dual and semi-dual objectives of our sparsity-constrained OT. Results are provided in Figure 3. Compared to ADAM, LBFGS is a more convenient option as it does not require the tuning of a learning rate hyperparameter. In addition, LBFGS empirically converges faster than ADAM in the number of iterations (first row of Figure 3). That being said, when a proper learning rate is chosen, we find that ADAM converges either as fast as or faster than LBFGS in wallclock

| | V-MoE B/32 | | V-MoE B/16 | |
|---|---|---|---|---|
| | JFT prec@1 | ImageNet 10 shots | JFT prec@1 | ImageNet 10 shots |
| TopK (Riquelme et al., 2021) | 43.47 | 65.91 | 48.86 | 72.12 |
| S-BASE (Clark et al., 2022) | 44.26 | 65.87 | 49.80 | 72.26 |
| SPARSITY-CONSTRAINED (ours) | **44.30** | **66.52** | **50.29** | **72.76** |

Table 2: **Performance of the V-MoE B/32 and B/16 architectures with different routers.** The fewshot experiments are averaged over 5 different seeds (Appendix A.5).

time (second row of Figure 3). In addition, Figure 3 shows that dual and semi-dual objectives are very close to each other toward the end of the optimization process. This empirically confirms Proposition 3, which states that the dual and the semi-dual are equal at their optimum.

We have seen that a greater $k$ leads to a smoother objective landscape (Figure 2). It is known that a smoother objective theoretically allows faster convergence. We validate this empirically in Appendix A.2, where we see that a greater $k$ leads to faster convergence.

### 6.2 SPARSE MIXTURES OF EXPERTS

We applied sparsity-constrained OT to vision sparse mixtures of experts (V-MoE) models for large-scale image recognition (Riquelme et al., 2021). A V-MoE model replaces a few dense feedforward layers MLP : $\boldsymbol{x} \in \mathbb{R}^d \mapsto \text{MLP}(\boldsymbol{x}) \in \mathbb{R}^d$ in Vision Transformer (ViT) (Dosovitskiy et al., 2021) with the sparsely-gated mixture-of-experts layers:

$$\text{MoE}(\boldsymbol{x}) \coloneqq \sum_{r=1}^{n} \text{Gate}_r(\boldsymbol{x}) \cdot \text{MLP}_r(\boldsymbol{x}), \tag{14}$$

where $\text{Gate}_r : \mathbb{R}^d \to \mathbb{R}_+$ is a **sparse gating function** and feedforward layers $\{\text{MLP}_r\}_{r=1}^n$ are **experts**. In practice, only those experts $\text{MLP}_r(\cdot)$ corresponding to a nonzero gate value $\text{Gate}_r(\boldsymbol{x})$ will be computed – in this case, we say that the token $\boldsymbol{x}$ is **routed** to the expert $r$. Upon a minibatch of $m$ tokens $\{\boldsymbol{x}_1, \ldots, \boldsymbol{x}_m\}$, we apply our sparsity-constrained OT to match tokens with experts, so that the number of tokens routed to any expert is bounded. Following Clark et al. (2022), we backprop the gradient only through the combining weights $\text{Gate}_r(\boldsymbol{x})$, but not through the OT algorithm (details in Appendix A.5), as this strategy accelerates the backward pass of V-MoEs. Using this routing strategy, we train the B/32 and B/16 variants of the V-MoE model: They refer to the "Base" variants of V-MoE with $32 \times 32$ patches and $16 \times 16$ patches, respectively. Hyperparameters of these architectures are provided in Riquelme et al. (2021, Appendix B.5). We train on the JFT-300M dataset (Sun et al., 2017), which is a large scale dataset that contains more than 305 million images. We then perform 10-shot transfer learning on the ImageNet dataset (Deng et al., 2009). Additional V-MoE and experimental details in Appendix A.5. Table 2 summarizes the validation accuracy on JFT-300M and 10-shot accuracy on ImageNet. Compared to baseline methods, our sparsity-constrained approach yields the highest accuracy with both architectures on both benchmarks.

## 7 CONCLUSION

We presented a dual and semi-dual framework for OT with general nonconvex regularization. We applied that framework to obtain a tractable lower bound to approximately solve an OT formulation with cardinality constraints on the columns of the transportation plan. We showed that this framework is formally equivalent to using squared $k$-support norm regularization in the primal. Moreover, it interpolates between unregularized OT (recovered when $k$ is small enough) and quadratically-regularized OT (recovered when $k$ is large enough). The (semi) dual objectives were shown to be increasingly smooth as $k$ increases, enabling the use of gradient-based algorithms such as LBFGS or ADAM. We illustrated our framework on a variety of tasks; see §6 and Appendix A. For training of mixture-of-experts models in large-scale computer vision tasks, we showed that a direct control of sparsity improves the accuracy, compared to top-k and Sinkhorn baselines. Beyond empirical performance, sparsity constraints may lead to more interpretable transportation plans and the integer-valued hyper-parameter $k$ may be easier to tune than the real-valued parameter $\gamma$.

Acknowledgments

We thank Carlos Riquelme, Antoine Rolet and Vlad Niculae for feedback on a draft of this paper, as well as Aidan Clark and Diego de Las Casas for discussions on the Sinkhorn-Base router. We are also grateful to Basil Mustafa, Rodolphe Jenatton, André Susano Pinto and Neil Houlsby for feedback throughout the project regarding MoE experiments. We thank Ryoma Sato for a fruitful email exchange regarding strong duality and ties.

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

# Appendix

|  | Regularization | Transportation plan | Preferred algorithm |
|---|---|---|---|
| Unregularized | None | Sparse | Network flow |
| Entropy-regularized | Convex | Dense | Sinkhorn |
| Quadratic-regularized | Convex | Sparse | LBFGS / ADAM |
| Sparsity-constrained | Non-Convex | Sparse & cardinality-constrained | LBFGS / ADAM |

Table 3: Overview of how our method compares to others. Our method can be thought as a middle ground between unregularized OT and quadratically-regularized OT.

## A    EXPERIMENTAL DETAILS AND ADDITIONAL EXPERIMENTS

### A.1    ILLUSTRATIONS

**OT between 2D points.**    In Figure 1, we visualize the transportation plans between 2D points. These transportation plans are obtained based on different formulations of optimal transport, whose properties we recall in Table 3. The details of this experiment are as follows. We draw 20 samples from a Gaussian distribution $\mathcal{N}(\left[\begin{smallmatrix}0\\0\end{smallmatrix}\right], \left[\begin{smallmatrix}1&0\\0&1\end{smallmatrix}\right])$ as source points; we draw 20 samples from a different Gaussian distribution $\mathcal{N}(\left[\begin{smallmatrix}4\\4\end{smallmatrix}\right], \left[\begin{smallmatrix}1&-0.8\\-0.6&1\end{smallmatrix}\right])$ as target points. The cost matrix in $\mathbb{R}^{20\times20}$ contains the Euclidean distances between source and target points. The source and target marginals $\boldsymbol{a}$ and $\boldsymbol{b}$ are both probability vectors filled with values $1/20$. On the top row of Figure 1, blue lines linking source points and target points indicate nonzero values in the transportation plan obtained from each optimal transport formulation. These transportation plans are shown in the second row. Figure 1 confirms that, by varying $k$ in our sparsity-constrained formulation, we control the columnwise sparsity of the transportation plan.

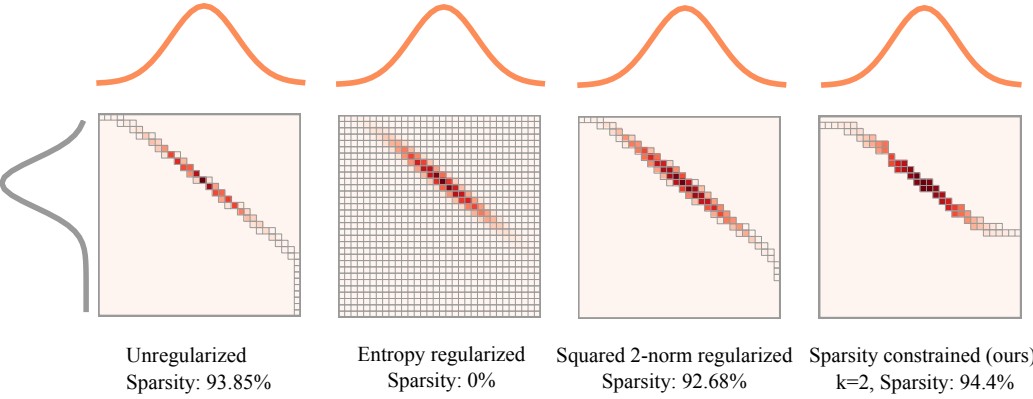

| Unregularized | Entropy regularized | Squared 2-norm regularized | Sparsity constrained (ours) |
|---|---|---|---|
| Sparsity: 93.85% | Sparsity: 0% | Sparsity: 92.68% | k=2, Sparsity: 94.4% |

Figure 4: OT between two Gaussian distributions.

**OT between two Gaussians.**    In Figure 4 above, we show transportation plans between two Gaussian distributions. The concrete set up of this experiment is as follows. We let $Y$ and $Z$ be categorical random variables taking values in $\{0, \dots, 31\}$. The realizations of $Y$ and $Z$ are the source and target locations on a 1D grid, respectively. Let $\phi(z; m, s) \coloneqq \exp\left(\frac{-(z-m)^2}{2s^2}\right)$, where $z, m, s$ are all real scalars with $s \neq 0$. The source distribution is set to be $\mathbb{P}(Y = y) \coloneqq \phi(y; 10, 4)/c_Y$ with a normalizing constant $c_Y \coloneqq \sum_{y=0}^{31} \phi(y; 10, 4)$; the target distribution is set to be $\mathbb{P}(Z = z) \coloneqq \phi(z; 16, 5)/c_Z$ with a normalizing constant $c_Z \coloneqq \sum_{z=0}^{31} \phi(z; 16, 5)$. The cost matrix $C$ contains normalized squared Euclidean distances between source and target locations: $C_{ij} = (i - j)^2/31^2 \in [0, 1]$. By setting $k = 2$ in our sparsity-constrained OT formulation, we obtain a transportation plan that contains at most two nonzeros per column (right-most panel of Figure 4).

**OT between Gaussian and bi-Gaussian.** Similar to Figure 4, we show transportation plans between a Gaussian source marginal and a mixture of two Gaussians target marginal in Figure 5. We set the source distribution as $\mathbb{P}(Y = y) := \phi(y; 16, 5)/c_Y$, where $c_Y := \sum_{y=0}^{31} \phi(y; 16, 5)$ is the normalizing constant; we set the target distribution as $\mathbb{P}(Z = z) := \big(\phi(z; 8, 5) + \phi(z; 24, 5)\big)/c_Z$, where $c_Z = \sum_{z=0}^{31} \big(\phi(z; 8, 5) + \phi(z; 24, 5)\big)$ is the normalizing constant. Apart from that, we use the same settings as Figure 4,

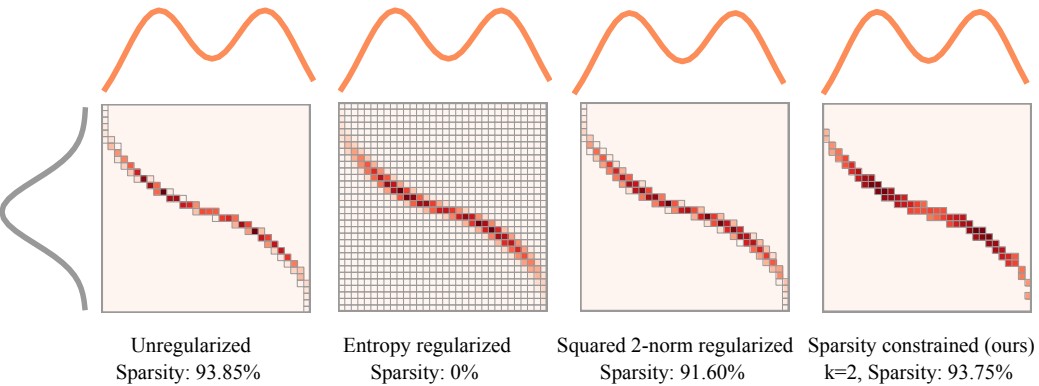

| Unregularized | Entropy regularized | Squared 2-norm regularized | Sparsity constrained (ours) |
|---|---|---|---|
| Sparsity: 93.85% | Sparsity: 0% | Sparsity: 91.60% | k=2, Sparsity: 93.75% |

Figure 5: OT between Gaussian and bi-Gaussian distributions.

## A.2 SOLVER COMPARISON WITH AN INCREASED CARDINALITY

We have seen that an increased $k$ increases the smoothness of the optimization problem (Figure 2). This suggests that solvers may converge faster with an increased $k$. We show this empirically in Figure 6, where we measure the gradient norm at each iteration of the solver and compare the case $k = 2$ and $k = 4$.

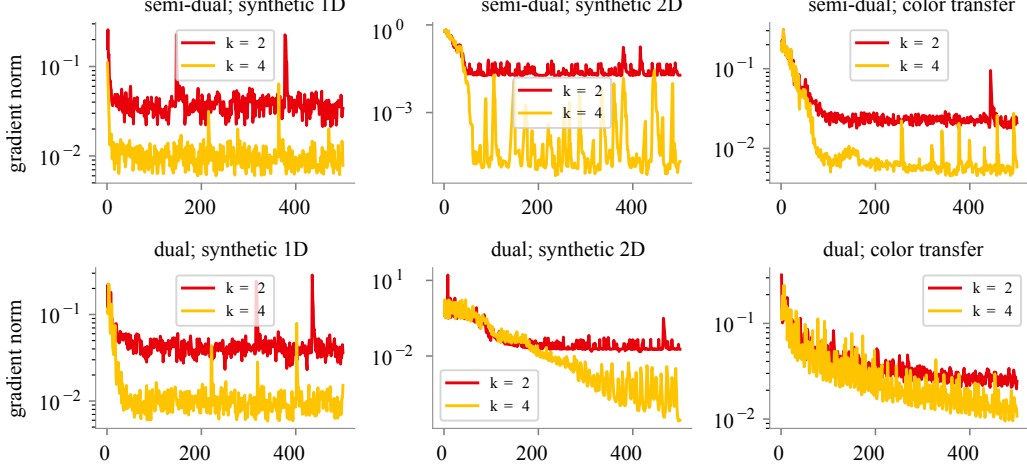

Figure 6: **Solvers converge faster with an increased** $k$**.** We measure the gradient norm at each iteration of LBFGS applied to the semi-dual formulations (top row) and the dual formulations (bottom row) of different datasets. Since the gradient norm should go to zero, we see that LBFGS solver converges faster with an increased $k$.

## A.3 COLOR TRANSFER

We apply our sparsity-constrained formulation on the classical OT application of color transfer (Pitié et al., 2007). We follow exactly the same experimental setup as in Blondel et al. (2018, Section 6).

Figure 7 shows the results obtained from our sparsity-constrained approach. Similar to well-studied alternatives, our yields visually pleasing results.

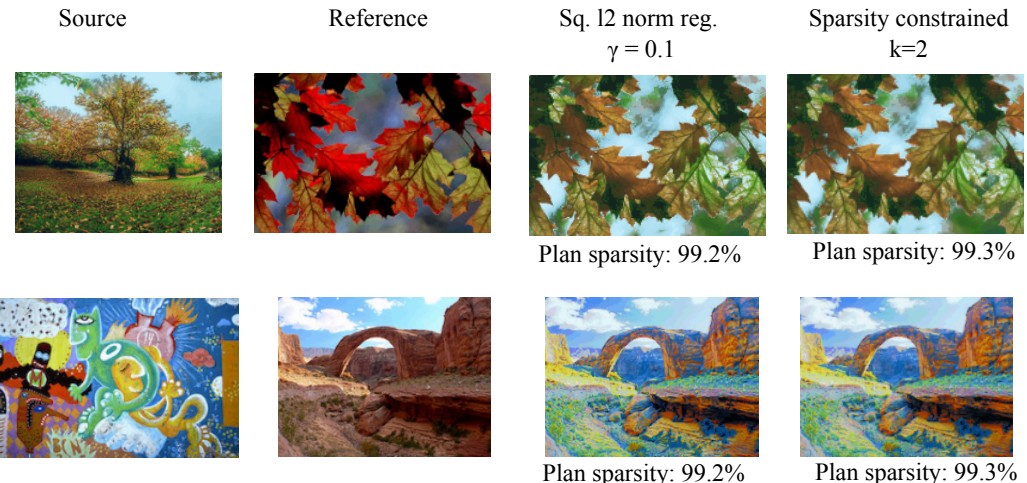

Figure 7: **Result comparison on the color transfer task**. The sparsity indicated below each image shows the percentage of nonzeros in the transportation plan. For a fair comparison, we use $k = 2$ for the sparsity-constrained formulation and the regularization weight $\gamma = 0.1$ for squared $\ell_2$ formulation to produce comparably sparse transportation plan.

## A.4    SUPPLY-DEMAND TRANSPORTATION ON SPHERICAL DATA

We follow Amos et al. (2022) to set up a synthetic transport problem between 100 supply locations and 10,000 demand locations worldwide. Transport costs are set to be the spherical distance between the demand and supply locations. This transportation problem can be solved via the entropy regularized optimal transport as in Amos et al. (2022). We visualized this entropy-regularized transport plan in panel **(a)** of Figure 8.

Building upon the setting in Amos et al. (2022), we additionally assume that each supplier has a **limited supplying capacity**. That is, each supplier can transport goods to as many locations as possible up to a certain prescribed limit. This constraint is conceivable, for instance, when suppliers operate with a limited workforce and cannot meet all requested orders. We incorporate this constraint into our formulation of sparsity-constrained optimal transport by specifying $k$ as the capacity limit. The panel **(b)** of Figure 8 is the obtained transportation plan with a supplying capacity of $k = 100$ (each supplier can transport goods to at most 100 demand locations).

Comparing panels **(a)** and **(b)** of Figure 8, we recognize that derived plans are visibly different in a few ways. For instance, with the capacity constraint on suppliers, demand locations in Europe import goods from more supply locations in North America than without the capacity constraint. Similar observations go to demand locations in pacific islands: Without the capacity constraint, demand locations in Pacific islands mostly rely on suppliers in North America; with the capacity constraint, additional suppliers in South America are in use.

**(a)** Entropy-regularized transportation plan

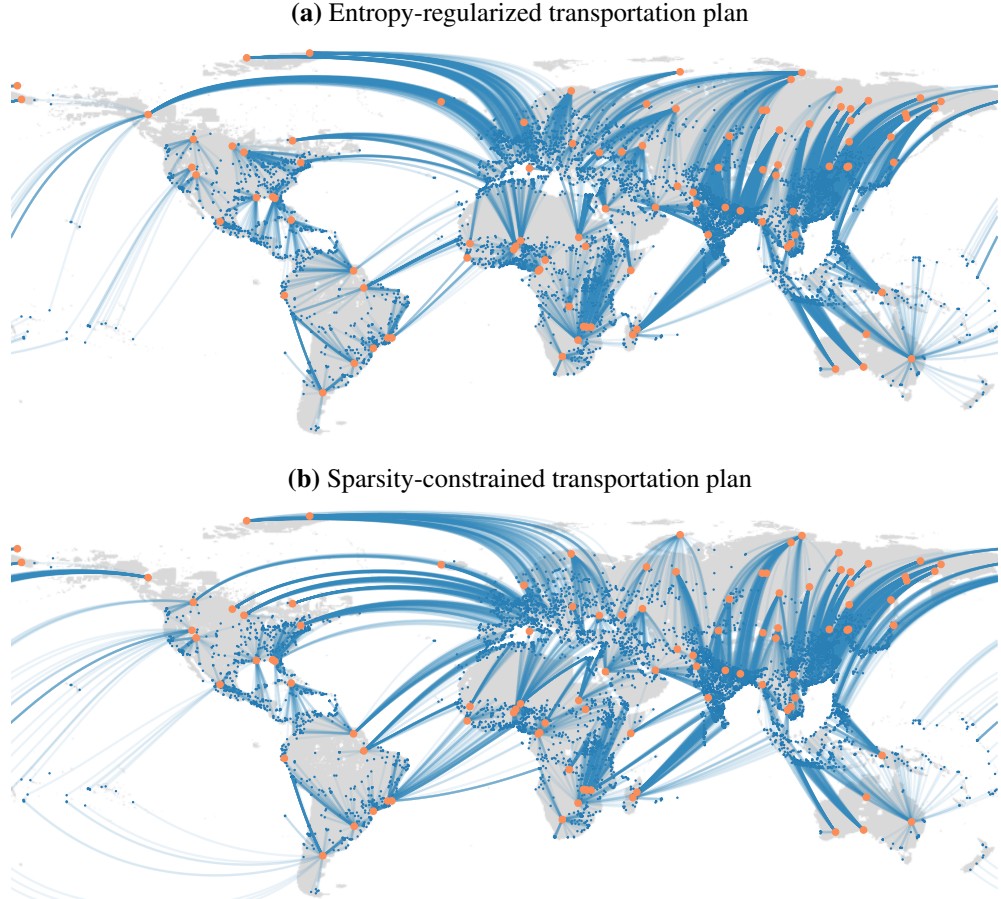

**(b)** Sparsity-constrained transportation plan

Figure 8: **Plans obtained from the supply-demand transportation task.** Blue lines show the transportation plan from the supply locations (yellow dots) to demand locations (blue dots). The top panel shows the transportation plan without a capacity limit on supply: Each supplying location can transport to as many demand location as possible. This is derived based on the entropy-regularized optimal transportation. The bottom panel shows the transportation plan with a capacity limit on supply: Each supplying location can meet demands up to a fixed capacity. The plan in this case is derived by sparsity-constrained optimal transport with $k = 100$.

A.5    V-MOE EXPERIMENT

Our experiment is based on the vision MoE (V-MoE) architecture (Riquelme et al., 2021), which replaces a few MLP layers of the vision Transformer (Dosovitskiy et al., 2021) by MoE layers. In this subsection, we review the background of V-MoE, with a focus on the **router**, which decides which experts get which input tokens.

We introduce a few notations that will be used throughout this subsection. Let $\{\boldsymbol{x}_1, \ldots, \boldsymbol{x}_m\} \subset \mathbb{R}^d$ be a minibatch of $m$ tokens in $\mathbb{R}^d$, and let $X \in \mathbb{R}^{m \times d}$ be a corresponding matrix whose rows are tokens. Let $W \in \mathbb{R}^{d \times n}$ be a learnable matrix of **expert weights**, where each column of $W$ is a learnable feature vector of an expert. Common to different routing mechanisms is an **token-expert affinity matrix** $\Pi \coloneqq XW \in \mathbb{R}^{m \times n}$: Its $(i, j)$-th entry is an inner-product similarity score between the $i$-th token and the $j$-th expert.

**The TopK router.**    To route tokens to experts, the **TopK router** in Riquelme et al. (2021) computes a sparse gating matrix $\Gamma$ that has at most $\kappa$ nonzeros per row, through a function $\mathrm{top}_\kappa : \mathbb{R}^n \to \mathbb{R}^n$ that sets all but largest $\kappa$ values zero:

$$\Gamma \coloneqq \mathrm{top}_\kappa\big(\mathrm{softmax}\,(\Pi + \sigma\boldsymbol{\epsilon})\big) \in \mathbb{R}^{m \times n} \quad \text{with} \quad \Pi = XW. \tag{15}$$

Note that the integer $\kappa$ is not to be confused with $k$ used in the main text – $\kappa$ here refers to the number of selected expert for each token and it can differ from the cardinality-constraint $k$ used in the main text in general. The vector $\boldsymbol{\epsilon} \sim \mathcal{N}(0, I)$ in (15) is a noise injected to the token-expert affinity matrix $XW$ with $\sigma \in \mathbb{R}$ controlling the strength of noise. In practice, $\sigma$ is set to be $1/n$ during training and 0 in inference. To ensure that all experts are sufficiently trained, the gating matrix $\Gamma$ in (15) is regularized by auxiliary losses that encourage experts to taken a similar amount of tokens in a minibatch. A detailed description of these auxiliary losses is presented in Riquelme et al. (2021, Section A).

For an efficient hardware utilization, Riquelme et al. (2021) allocate a **buffer capacity** of experts, which specifies the number of tokens each expert can **at most** process in a minibatch. With a specified buffer capacity and a computed gating matrix $\Gamma$, the TopK router goes over the rows of $\Gamma$ and assign each token to its top-chosen expert as long as the chosen expert's capacity is not full. This procedure is described in Algorithm 1 of Riquelme et al. (2021, Section C.1). Finally, the outcomes of experts are linearly combined using the gating matrix $\Gamma$ as in (14).

**The S-BASE router.**    Clark et al. (2022) cast the token-expert matching problem as an entropy-regularized OT problem, solved using the Sinkhorn algorithm. This approach, dubbed as the Sinkhorn-BASE (S-BASE) router, was originally designed for language MoEs that take text as input. In this work, we adapt it to vision MoEs. In direct parallel to the TopK gating matrix in (15), the gating matrix of entropy-regularized OT is set to be

$$\Gamma_{\mathrm{ent}} \coloneqq \mathrm{top}_\kappa\big(\Pi_{\mathrm{ent}}\big) \in \mathbb{R}^{m \times n}, \tag{16}$$

where

$$\Pi_{\mathrm{ent}} \coloneqq \underset{T \in \mathcal{U}(\boldsymbol{a}, \boldsymbol{b})}{\mathrm{argmin}} \, \langle T, -\Pi \rangle + \langle T, \log T \rangle, \quad \text{with } \boldsymbol{a} = \mathbf{1}_n \text{ and } \boldsymbol{b} = (m/n)\mathbf{1}_n. \tag{17}$$

The optimization plan $\Pi_{\mathrm{ent}}$ in (17) can be obtained using the Sinkhorn algorithm (Sinkhorn & Knopp, 1967). Note that while we formulated optimal transport problems with non-negative cost matrices in the main text, values in the cost matrix $C = -\Pi$ in (17) can be both positive and negative, following Clark et al. (2022). Since $\Pi_{\mathrm{ent}}$ is a dense matrix, a heuristic is needed to select only $\kappa$ experts to form the gating matrix $\Gamma_{\mathrm{ent}}$ – this is achieved by using a $\mathrm{top}_\kappa$ in (16). With a computed gating matrix $\Gamma_{\mathrm{ent}}$, the S-BASE router assigns each token to its top-chosen expert in the same way of the TopK router. This process allocates each expert an amount of tokens, up to a certain upper bound specified by the buffer capacity as in the case of TopK. As in Clark et al. (2022), we linearly combine the output of experts using a softmax matrix $\mathrm{softmax}(\Pi)$. In this way, the backward pass of gradient-based training does not go through the Sinkhorn algorithm, can be faster and more numerically stable[1].

---

[1]Personal communications with the authors of Clark et al. (2022).

**The Sparsity-constrained router.**  We cast the token-expert matching problem as a sparsity-constrained OT problem. With a prescribed buffer capacity $k$, our goal is to upper-bound the number of tokens assigned to each expert by $k$. This amounts to adding a cardinality constraint to each column of the gating matrix:

$$\Gamma_{\text{sparse}} := \underset{\substack{T \in \mathcal{U}(\boldsymbol{a}, \boldsymbol{b}) \\ T \in \mathcal{B}_k \times \cdots \times \mathcal{B}_k}}{\operatorname{argmin}} \langle T, -\Pi_{\text{softmax}} \rangle + \frac{1}{2} \|T\|_2^2, \quad \text{with } \boldsymbol{a} = \mathbf{1}_n \text{ and } \boldsymbol{b} = (m/n)\mathbf{1}_n \quad (18)$$

with $\Pi_{\text{softmax}} = \text{softmax}(XW)$. The purpose of the softmax function here is to obtain a cost matrix containing values of the same sign. Otherwise, if a cost matrix contains both positive and negative values, then the obtained plan from sparsity-constrained optimal transport may contain zero at all entries corresponding to positive values in the cost matrix, so as to minimize to the objective. In that case, columns of this transportation may contain much fewer nonzeros than $k$ – this is an undesirable situation as it under-uses the buffer capacity of experts. Note that, however, this was not an issue in the S-BASE router – a cost matrix there can contain both positive and negative values (Clark et al., 2022) – because values of the transportation plan yielded by the Sinkhorn's algorithm are strictly positive.

The sparse transportation plan $\Gamma_{\text{sparse}}$ in (18), allocates each expert an amount of tokens up to $k$. As in the S-BASE router, we linearly combine the output of experts using the matrix $\Pi_{\text{softmax}}$.

To approximate $\Gamma_{\text{sparse}}$, we optimize its semi-dual proxy as introduced in Section 4. We do so by using an ADAM optimizer with a learning rate of $10^{-2}$ for 50 steps.

**V-MoE architecture.**  We use the S-BASE router and our proposed sparsity-constrained router as drop-in replacements of the TopK router in otherwise standard V-MoE architectures (Riquelme et al., 2021). We focus on the V-MoE B/32 and B/16 architectures, which use $32 \times 32$ and $16 \times 16$ patches, respectively. We place MoEs on every other layer, which is the Every-2 variant in Riquelme et al. (2021). We fix the total number of experts $n = 32$ for all experiments. In the TopK and S-BASE router, we assign 2 experts to each expert, that is, $\kappa = 2$ in (15) and (16). The buffer capacity is set to be $n/\kappa = 32/2 = 16$, that is, each expert can take 16 tokens at most. To match this setting, we use $k = 16$ in (18) for our sparsity-constrained router.

**Upstream training and evaluation.**  We follow the same training strategy of Riquelme et al. (2021) to train B/32 and B/16 models on JFT-300M, with hyperparameters reported in Riquelme et al. (2021, Table 8). JFT-300M has around 305M training and 50,000 validation images. Since labels of the JFT-300M are organized in a hierarchical way, an image may associate with multiple labels. We report the model performance by precision@1 by checking if the predicted class with the highest probability is one of the true labels of the image.

**Downstream transfer to ImageNet.**  For downstream evaluations, we perform 10-shot linear transfer on ImageNet (Deng et al., 2009). Specifically, with a JFT-trained V-MoE model, we freeze the model up to its penultimate layer, re-initialize its last layer, and train the last layer on ImageNet. This newly initialized layer is trained on 10 examples per ImageNet class (10-shot learning).

|  | B/32 | B/16 |
|---|---|---|
| TopK (Riquelme et al., 2021) | 97.11 | 308.14 |
| S-BASE (Clark et al., 2022) | 98.88 | 312.52 |
| Sparsity-constrained (ours) | 122.66 | 433.56 |

Table 4: Total Training TPUv2-core-days

**Comparing the speed of routers.**  We note that the sparsity-constrained router is slightly slower than baseline routers. One reason is that the topk function used for $k$-sparse projection steps. To further speedup the sparsity-constrained router, an option is to use the approximated version of topk (Chern et al., 2022), which we did not use in this study. This approximated topk may be especially useful on large models like B/16, where the number of tokens is large. Another way to accelerate

|  | E-Step | M-Step |
|---|---|---|
| K-Means | $t_{i,:} \leftarrow e_{\mathrm{argmin}_j [C^{(s)}]_{i,j}}$ | |
| Soft K-Means | $t_{i,:} \leftarrow \mathrm{softmax}\big( - [C^{(s)}]_{i,:} \big)$ | $\mu^{(s+1)} \leftarrow (XT) \oslash (\mathbf{1}_{d \times m} T)$ |
| Negentropy | Solve (1) with $\Omega(\boldsymbol{t}) = \langle \boldsymbol{t}, \log \boldsymbol{t} \rangle$ | |
| Squared 2-norm | Solve (1) with $\Omega(\boldsymbol{t}) = \frac{1}{2}\|\boldsymbol{t}\|_2^2$ | |
| Sparsity-constrained | Solve (4), (5) with $\Omega(\boldsymbol{t}) = \frac{1}{2}\|\boldsymbol{t}\|_2^2 + \delta_{\mathcal{B}_k}(\boldsymbol{t})$ | |

Table 5: Steps of the EM-like algorithm used to estimate the centers of the clusters with each method. In all cases, the cost matrix $C^{(s)}$ is the squared distance between each data point and the current estimate of the centers, at a given step $s$, i.e. $[C^{(s)}]_{i,j} = \|\boldsymbol{x}_i - \boldsymbol{\mu}_j^{(s)}\|_2^2$. The vector $\boldsymbol{e}_l \in \mathbb{R}^n$ is a canonical basis vector with $l$-th entry being 1 and all other entries being 0.

the sparsity-constrained router is to explore different optimizers. Currently, we run the ADAM optimizer for 50 steps using a learning rate $10^{-2}$. We suspect that with a more careful tuning of the optimizer, one can reduce the number of steps without harming the performance. Variants of accelerated gradient-based methods (An et al., 2022) may also be applicable.

### A.6 SOFT BALANCED CLUSTERING

**OT viewpoint.** Suppose we want to cluster $m$ data points $\boldsymbol{x}_1, ..., \boldsymbol{x}_m \in \mathbb{R}^d$ into $n$ clusters with centroids $\boldsymbol{\mu}_1, \ldots, \boldsymbol{\mu}_n \in \mathbb{R}^d$. We let $X \in \mathbb{R}^{d \times m}$ be a matrix that contains data points $\boldsymbol{x}_1, ..., \boldsymbol{x}_m$ as columns. Similarly, we let $\mu \in \mathbb{R}^{d \times n}$ be a matrix of centroids.

The K-Means algorithm can be viewed as an OT problem with only one marginal constraint,

$$\min_{\substack{T \in \mathbb{R}_+^{m \times n} \\ T\mathbf{1}_n = \boldsymbol{a} \\ \mu \in \mathbb{R}^{d \times n}}} \sum_{i=1}^m \sum_{j=1}^n t_{i,j}\|\boldsymbol{x}_i - \boldsymbol{\mu}_j\|_2^2 = \min_{\substack{T \in \mathbb{R}_+^{m \times n} \\ T\mathbf{1}_n = \boldsymbol{a} \\ \mu \in \mathbb{R}^{d \times n}}} \langle T, C \rangle,$$

where $[C]_{i,j} = \|\boldsymbol{x}_i - \boldsymbol{\mu}_j\|_2^2$ and $\boldsymbol{a} = \mathbf{1}_m/m$. Lloyd's algorithm corresponds to alternating minimization w.r.t. $T$ (updating centroid memberships) and w.r.t. $\mu$ (updating centroid positions).

This viewpoint suggests two generalizations. The first one consists in using two marginal constraints

$$\min_{\substack{T \in \mathbb{R}_+^{m \times n} \\ T\mathbf{1}_n = \boldsymbol{a} \\ T^\top \mathbf{1}_m = \boldsymbol{b} \\ \mu \in \mathbb{R}^{d \times n}}} \langle T, C \rangle = \min_{\substack{T \in \mathcal{U}(\boldsymbol{a}, \boldsymbol{b}) \\ \mu \in \mathbb{R}^{d \times n}}} \langle T, C \rangle.$$

This is useful in certain applications to impose a prescribed size to each cluster (e.g., $\boldsymbol{b} = \mathbf{1}_n/n$) and is sometimes known as balanced or constrained K-Means (Ng, 2000).

The second generalization consists in introducing convex regularization $\Omega$

$$\min_{\substack{T \in \mathcal{U}(\boldsymbol{a}, \boldsymbol{b}) \\ \mu \in \mathbb{R}^{d \times n}}} \langle T, C \rangle + \sum_{j=1}^n \Omega(\boldsymbol{t}_j).$$

This moves the optimal plan away from the vertices of the polytope. This corresponds to a "soft" balanced K-Means, in which we replace "hard" cluster memberships with "soft" ones. We can again alternate between minimization w.r.t. $T$ (solving a regularized OT problem) and minimization w.r.t. $\mu$. In the case of the squared Euclidean distance, the closed form solution for the latter is $\boldsymbol{\mu}_i \propto \sum_{j=1}^n t_{i,j}\boldsymbol{x}_j$ for all $i \in [m]$.

When $\Omega$ is nonconvex, we propose to solve the (semi) dual as discussed in the main text.

**Results on MNIST.** MNIST contains grayscale images of handwritten digits, with a resolution of $28 \times 28$ pixels. The dataset is split in $60\,000$ training and $10\,000$ test images. As preprocessing,

|  | Cost | KL |
|---|---|---|
| K-Means | $161.43 \pm 2.20$ | $0.227502 \pm 0.094380$ |
| Soft K-Means | $\mathbf{157.10 \pm 0.19}$ | $0.040756 \pm 0.007071$ |
| Negentropy | $157.45 \pm 0.04$ | $0.000029 \pm 0.000002$ |
| Squared 2-norm | $157.63 \pm 0.26$ | $0.000018 \pm 0.000001$ |
| Sparsity-constrained | $157.53 \pm 0.06$ | $\mathbf{0.000013 \pm 0.000001}$ |

Table 6: Clustering results on MNIST using different algorithms. We report the average cost on the test split (average distance between each test image and its cluster), and the Kullback–Leibler divergence between the empirical distribution of images per cluster and the expected one (a uniform distribution). We average the results over 20 runs and report confidence intervals at 95%. The algorithm proposed in §5 achieves the most balanced clustering, and a comparable cost to other OT-based solutions.

we simply put the pixel values in the range $[-1, 1]$ and "flatten" the images to obtain vectors of 784 elements.

We use the training set to estimate the centers of the clusters using different algorithms. We use an EM-like algorithm to estimate the cluster centers in all cases, as described in Table 5 (we perform 50 update steps). In particular, notice that only the E-step changes across different algorithms, as described in Table 5. Since there are 10 digits, we use 10 clusters.

We evaluate the performance on the test set. Since some of the algorithms produce a "soft" clustering (all except K-Means), represented by the matrix $T$, for each test image $i$ we assign it to the cluster $j$ with the largest value in $T = (t_{i,j})$. We measure the average cost (i.e. average squared distance between each image and its selected cluster), and the KL divergence between the empirical distribution of images per cluster and the expected one (a uniform distribution). The centers are initialized from a normal distribution with a mean of 0 and a standard deviation of $10^{-3}$. Algorithms employing an OT-based approach perform 500 iterations to find $T$, using either the Sinkhorn algorithm (with the Negentropy method) or LBFGS (used by the rest of OT-based methods). We use a sparsity-constraint of $k = \lceil 1.15 \cdot \frac{m}{n} \rceil$ (recall that $k$ is the maximum number of nonzeros per column). Notice that using $k = \frac{m}{n}$, and assuming that $n$ is a divisor of $m$, would necessary require that the number of nonzeros per row is 1. Thus, our minimization problem would be equivalent to that of the unregularized OT. Thus, we slightly soften the regularization.

Table 6 shows the results of the experiment, averaged over 20 different random seeds. The best cost is achieved by the Soft K-Means algorithm, but the resulting clustering is quite unbalanced, as reported by the KL divergence metric. On the other hand, all OT-based approaches achieve similar costs, but the algorithm based on §5 obtains a significantly better balanced clustering.

# B   PROOFS

## B.1   WEAK DUALITY (PROPOSITION 1)

Recall that

$$P_\Omega(\boldsymbol{a}, \boldsymbol{b}, C) = \min_{T \in \mathcal{U}(\boldsymbol{a},\boldsymbol{b})} \langle T, C \rangle + \sum_{j=1}^{n} \Omega(\boldsymbol{t}_j) = \min_{\substack{T \in \mathbb{R}_+^{m \times n} \\ T\mathbf{1}_n = \boldsymbol{a} \\ T^\top \mathbf{1}_m = \boldsymbol{b}}} \sum_{j=1}^{n} \langle \boldsymbol{t}_j, \boldsymbol{c}_j \rangle + \Omega(\boldsymbol{t}_j).$$

We add Lagrange multipliers for the two equality constraints but keep the constraint $T \in \mathbb{R}_+^{m \times n}$ explicitly. The Lagrangian is then

$$L(T, \boldsymbol{\alpha}, \boldsymbol{\beta}) \coloneqq \sum_{j=1}^{n} \langle \boldsymbol{t}_j, \boldsymbol{c}_j \rangle + \Omega(\boldsymbol{t}_j) - \langle \boldsymbol{\alpha}, T\mathbf{1}_n - \boldsymbol{a} \rangle - \langle \boldsymbol{\beta}, T^\top \mathbf{1}_m - \boldsymbol{b} \rangle$$

$$= \sum_{j=1}^{n} \langle \boldsymbol{t}_j, \boldsymbol{c}_j - \boldsymbol{\alpha} - \beta_j \mathbf{1}_m \rangle + \Omega(\boldsymbol{t}_j) + \langle \boldsymbol{\alpha}, \boldsymbol{a} \rangle + \langle \boldsymbol{\beta}, \boldsymbol{b} \rangle.$$

Using the inequality $\min_{\boldsymbol{u}} \max_{\boldsymbol{v}} f(\boldsymbol{u}, \boldsymbol{v}) \geq \max_{\boldsymbol{v}} \min_{\boldsymbol{u}} f(\boldsymbol{u}, \boldsymbol{v})$ twice, we have

$$
\begin{aligned}
P_\Omega(\boldsymbol{a}, \boldsymbol{b}, C) &= \min_{T \in \mathbb{R}_+^{m \times n}} \max_{\boldsymbol{\alpha} \in \mathbb{R}^m} \max_{\boldsymbol{\beta} \in \mathbb{R}^n} L(T, \boldsymbol{\alpha}, \boldsymbol{\beta}) \\
&\geq \max_{\boldsymbol{\alpha} \in \mathbb{R}^m} \min_{T \in \mathbb{R}_+^{m \times n}} \max_{\boldsymbol{\beta} \in \mathbb{R}^n} L(T, \boldsymbol{\alpha}, \boldsymbol{\beta}) \\
&\geq \max_{\boldsymbol{\alpha} \in \mathbb{R}^m} \max_{\boldsymbol{\beta} \in \mathbb{R}^n} \min_{T \in \mathbb{R}_+^{m \times n}} L(T, \boldsymbol{\alpha}, \boldsymbol{\beta}).
\end{aligned}
$$

For the first inequality, we have

$$
\max_{\boldsymbol{\alpha} \in \mathbb{R}^m} \min_{T \in \mathbb{R}_+^{m \times n}} \max_{\boldsymbol{\beta} \in \mathbb{R}^n} L(T, \boldsymbol{\alpha}, \boldsymbol{\beta}) = \max_{\boldsymbol{\alpha} \in \mathbb{R}^m} \langle \boldsymbol{\alpha}, \boldsymbol{a} \rangle + \min_{T \in \mathbb{R}_+^{m \times n}} \max_{\boldsymbol{\beta} \in \mathbb{R}^n} \langle \boldsymbol{\beta}, \boldsymbol{b} \rangle + \sum_{j=1}^n \langle \boldsymbol{t}_j, \boldsymbol{c}_j - \boldsymbol{\alpha} - \beta_j \mathbf{1}_m \rangle + \Omega(\boldsymbol{t}_j)
$$

$$
= \max_{\boldsymbol{\alpha} \in \mathbb{R}^m} \langle \boldsymbol{\alpha}, \boldsymbol{a} \rangle + \sum_{j=1}^n \min_{\boldsymbol{t}_j \in b_j \triangle^m} \langle \boldsymbol{t}_j, \boldsymbol{c}_j - \boldsymbol{\alpha} \rangle + \Omega(\boldsymbol{t}_j)
$$

$$
= \max_{\boldsymbol{\alpha} \in \mathbb{R}^m} \langle \boldsymbol{\alpha}, \boldsymbol{a} \rangle - \sum_{j=1}^n \Omega_{b_j}^*(\boldsymbol{\alpha} - \boldsymbol{c}_j)
$$

$$
= S_\Omega(\boldsymbol{a}, \boldsymbol{b}, C).
$$

For the second inequality, we have

$$
\max_{\boldsymbol{\alpha} \in \mathbb{R}^m} \max_{\boldsymbol{\beta} \in \mathbb{R}^n} \min_{T \in \mathbb{R}_+^{m \times n}} L(T, \boldsymbol{\alpha}, \boldsymbol{\beta}) = \max_{\boldsymbol{\alpha} \in \mathbb{R}^m} \max_{\boldsymbol{\beta} \in \mathbb{R}^n} \langle \boldsymbol{\alpha}, \boldsymbol{a} \rangle + \langle \boldsymbol{\beta}, \boldsymbol{b} \rangle + \sum_{j=1}^n \min_{\boldsymbol{t}_j \in \mathbb{R}_+^m} \langle \boldsymbol{t}_j, \boldsymbol{c}_j - \boldsymbol{\alpha} - \beta_j \mathbf{1}_m \rangle + \Omega(\boldsymbol{t}_j)
$$

$$
= \max_{\boldsymbol{\alpha} \in \mathbb{R}^m} \max_{\boldsymbol{\beta} \in \mathbb{R}^n} \langle \boldsymbol{\alpha}, \boldsymbol{a} \rangle + \langle \boldsymbol{\beta}, \boldsymbol{b} \rangle - \sum_{j=1}^n \Omega_+^*(\boldsymbol{\alpha} + \beta_j \mathbf{1}_m - \boldsymbol{c}_j)
$$

$$
= D_\Omega(\boldsymbol{a}, \boldsymbol{b}, C).
$$

To summarize, we showed $P_\Omega(\boldsymbol{a}, \boldsymbol{b}, C) \geq S_\Omega(\boldsymbol{a}, \boldsymbol{b}, C) \geq D_\Omega(\boldsymbol{a}, \boldsymbol{b}, C)$.

## B.2 DUAL-PRIMAL LINK

When the solution of the maximum below is unique, $\boldsymbol{t}_j^\star$ can be uniquely determined for $j \in [n]$ from

$$
\begin{aligned}
\boldsymbol{t}_j^\star &= \nabla \Omega_+^*(\boldsymbol{\alpha}^\star + \beta_j^\star \mathbf{1}_m - \boldsymbol{c}_j) = \underset{\boldsymbol{t}_j \in \mathbb{R}_+^M}{\operatorname{argmax}} \, \langle \boldsymbol{\alpha}^\star + \beta_j^\star \mathbf{1}_m - \boldsymbol{c}_j, \boldsymbol{t}_j \rangle - \Omega(\boldsymbol{t}_j) \\
&= \nabla \Omega_{b_j}^*(\boldsymbol{\alpha}^\star - \boldsymbol{c}_j) = \underset{\boldsymbol{t}_j \in \triangle^m}{\operatorname{argmax}} \, \langle \boldsymbol{\alpha}^\star - \boldsymbol{c}_j, \boldsymbol{t}_j \rangle - \Omega(\boldsymbol{t}_j).
\end{aligned} \tag{19}
$$

See Table 1 for examples. When the maximum is not unique, $\boldsymbol{t}_j^\star$ is jointly determined by

$$
\begin{aligned}
\boldsymbol{t}_j^\star &\in \partial \Omega_+^*(\boldsymbol{\alpha}^\star + \beta_j^\star \mathbf{1}_m - \boldsymbol{c}_j) = \underset{\boldsymbol{t}_j \in \mathbb{R}_+^M}{\operatorname{argmax}} \, \langle \boldsymbol{\alpha}^\star + \beta_j^\star \mathbf{1}_m - \boldsymbol{c}_j, \boldsymbol{t}_j \rangle - \Omega(\boldsymbol{t}_j) \\
&\in \partial \Omega_{b_j}^*(\boldsymbol{\alpha}^\star - \boldsymbol{c}_j) = \underset{\boldsymbol{t}_j \in \triangle^m}{\operatorname{argmax}} \, \langle \boldsymbol{\alpha}^\star - \boldsymbol{c}_j, \boldsymbol{t}_j \rangle - \Omega(\boldsymbol{t}_j),
\end{aligned}
$$

where $\partial$ indicates the subdifferential, and by the primal feasability $T^\star \in \mathcal{U}(\boldsymbol{a}, \boldsymbol{b})$, or more explicitly

$$
T^\star \in \mathbb{R}_+^{m \times n}, \quad T^\star \mathbf{1}_n = \boldsymbol{a} \quad \text{and} \quad (T^\star)^\top \mathbf{1}_m = \boldsymbol{b}.
$$

This also implies $\langle T^\star, \mathbf{1}_m \mathbf{1}_n^\top \rangle = 1$.

**Unregularized case.** When $\Omega = 0$, for the dual, we have

$$
\partial \Omega_+^*(\boldsymbol{s}_j) = \underset{\boldsymbol{t}_j \in \mathbb{R}_+^m}{\operatorname{argmax}} \, \langle \boldsymbol{s}_j, \boldsymbol{t}_j \rangle.
$$

We note that the problem is coordinate-wise separable with

$$\underset{t_{i,j} \in \mathbb{R}_+}{\operatorname{argmax}} \ t_{i,j} \cdot s_{i,j} = \begin{cases} \varnothing & \text{if } s_{i,j} > 0 \\ \mathbb{R}_{++} & \text{if } s_{i,j} = 0 \\ \{0\} & \text{if } s_{i,j} < 0 \end{cases}.$$

With $s_{i,j} = \alpha_i^\star + \beta_j^\star - c_{i,j}$, we therefore obtain

$$\begin{cases} t_{i,j}^\star > 0 & \text{if } \alpha_i^\star + \beta_j^\star = c_{i,j} \\ t_{i,j}^\star = 0 & \text{if } \alpha_i^\star + \beta_j^\star < c_{i,j} \end{cases},$$

since $s_{i,j} > 0$ is dual infeasible. We can therefore use $\boldsymbol{\alpha}^\star$ and $\boldsymbol{\beta}^\star$ to identify the support of $T^\star$. The size of that support is at most $m+n-1$ (Peyré & Cuturi, 2019, Proposition 3.4). Using the marginal constraints $T^\star \mathbf{1}_n = \boldsymbol{a}$ and $(T^\star)^\top \mathbf{1}_m = \boldsymbol{b}$, we can therefore form a system of linear equations of size $m + n$ to recover $T^\star$.

Likewise, for the semi-dual, with $\boldsymbol{s}_j = \boldsymbol{\alpha}^\star - \boldsymbol{c}_j$, we have

$$\begin{aligned}
\boldsymbol{t}_j^\star &\in \partial \Omega_{b_j}^*(\boldsymbol{s}_j) \\
&= \underset{\boldsymbol{t}_j \in \triangle^m}{\operatorname{argmax}} \ \langle \boldsymbol{s}_j, \boldsymbol{t}_j \rangle \\
&= \operatorname{conv}(\{\boldsymbol{v}_1, \dots, \boldsymbol{v}_{|S_j|}\}) \\
&= \operatorname{conv}(S_j),
\end{aligned}$$

where $S_j := \operatorname{conv}(\{\boldsymbol{e}_i : i \in \operatorname{argmax}_{i \in [m]} s_{i,j}\})$. Let us gather $\boldsymbol{v}_1, \dots, \boldsymbol{v}_{|S_j|}$ as a matrix $V_j \in \mathbb{R}^{m \times |S_j|}$. There exists $\boldsymbol{w}_j \in \triangle^{|S_j|}$ such that $\boldsymbol{t}_j^\star = V_j \boldsymbol{w}_j$. Using the primal feasability, we can solve with respect to $\boldsymbol{w}_j$ for $j \in [n]$. This leads to a (potentially undertermined) system of linear equations with $\sum_{j=1}^n |S_j|$ unknowns and $m + n$ equations.

**Squared $k$-support norm.** We now discuss $\Omega = \Psi$, as defined in (13). When the maximum is unique (no ties), $\boldsymbol{t}_j^\star$ is uniquely determined by (19). We now discuss the case of ties.

For the dual, with $\boldsymbol{s}_j = \boldsymbol{\alpha}^\star + \beta_j^\star \mathbf{1}_m - \boldsymbol{c}_j$, we have

$$\begin{aligned}
\boldsymbol{t}_j^\star &\in \partial \Omega_+^*(\boldsymbol{s}_j) \\
&= \underset{\boldsymbol{t}_j \in \mathbb{R}_+^m}{\operatorname{argmax}} \ \langle \boldsymbol{s}_j, \boldsymbol{t}_j \rangle - \Omega(\boldsymbol{t}_j) \\
&= \operatorname{conv}(\{\boldsymbol{v}_1, \dots, \boldsymbol{v}_{|S_j|}\}) \\
&:= \operatorname{conv}(\{[\boldsymbol{u}_j]_+ : \boldsymbol{u}_j \in S_j\}),
\end{aligned}$$

where $S_j := \operatorname{topk}(\boldsymbol{s}_j)$ is a set containing all possible top-$k$ vectors (the set is a singleton if there are no ties in $\boldsymbol{s}_j$, meaning that there is only one possible top-$k$ vector).

For the semi-dual, with $\boldsymbol{s}_j = \boldsymbol{\alpha}^\star - \boldsymbol{c}_j$, we have

$$\begin{aligned}
\boldsymbol{t}_j^\star &\in \partial \Omega_{b_j}^*(\boldsymbol{s}_j) \\
&= \underset{\boldsymbol{t}_j \in \triangle^m}{\operatorname{argmax}} \ \langle \boldsymbol{s}_j, \boldsymbol{t}_j \rangle - \Omega(\boldsymbol{t}_j) \\
&= \operatorname{conv}(\{\boldsymbol{v}_1, \dots, \boldsymbol{v}_{|S_j|}\}) \\
&:= \operatorname{conv}(\{[\boldsymbol{u}_j - \tau_j]_+ : \boldsymbol{u}_j \in S_j\}),
\end{aligned}$$

where $\tau_j$ is such that $\sum_{i=1}^k [s_{[i],j} - \tau_j]_+ = b_j$. Again, we can combine these conditions with the primal feasability $T^\star \in \mathcal{U}(\boldsymbol{a}, \boldsymbol{b})$ to obtain a system of linear equations. Unfortunately, in case of ties, ensuring that $T^\star \in \mathcal{U}(\boldsymbol{a}, \boldsymbol{b})$ by solving this system may cause $\boldsymbol{t}_j^\star \notin \mathcal{B}_k$. Another situation causing $\boldsymbol{t}_j^\star \notin \mathcal{B}_k$ is if $k$ is set to a smaller value than the maximum number of nonzero elements in the columns of the primal LP solution.

### B.3 PRIMAL INTERPRETATION (PROPOSITION 2)

For the semi-dual, we have

$$
\begin{aligned}
S_\Omega(\boldsymbol{a}, \boldsymbol{b}, C) &= \max_{\boldsymbol{\alpha} \in \mathbb{R}^m} \langle \boldsymbol{\alpha}, \boldsymbol{a} \rangle - \sum_{j=1}^n \Omega_{b_j}^*(\boldsymbol{\alpha} - \boldsymbol{c}_j) \\
&= \max_{\substack{\boldsymbol{\alpha} \in \mathbb{R}^m \\ \boldsymbol{\mu}_j : \, \boldsymbol{\mu}_j = \boldsymbol{\alpha} - \boldsymbol{c}_j}} \langle \boldsymbol{\alpha}, \boldsymbol{a} \rangle - \sum_{j=1}^n \Omega_{b_j}^*(\boldsymbol{\mu}_j) \\
&= \max_{\substack{\boldsymbol{\alpha} \in \mathbb{R}^m \\ \boldsymbol{\mu}_j \in \mathbb{R}^m}} \min_{T \in \mathbb{R}^{m \times n}} \langle \boldsymbol{\alpha}, \boldsymbol{a} \rangle - \sum_{j=1}^n \Omega_{b_j}^*(\boldsymbol{\mu}_j) + \sum_{j=1}^n \langle \boldsymbol{t}_j, \boldsymbol{\mu}_j - \boldsymbol{\alpha} + \boldsymbol{c}_j \rangle \\
&= \min_{T \in \mathbb{R}^{m \times n}} \langle T, C \rangle + \max_{\boldsymbol{\alpha} \in \mathbb{R}^m} \langle \boldsymbol{\alpha}, T \mathbf{1}_n - \boldsymbol{a} \rangle + \sum_{j=1}^n \max_{\boldsymbol{\mu}_j \in \mathbb{R}^m} \langle \boldsymbol{t}_j, \boldsymbol{\mu}_j \rangle - \Omega_{b_j}^*(\boldsymbol{\mu}_j) \\
&= \min_{\substack{T \in \mathbb{R}^{m \times n} \\ T \mathbf{1}_n = \boldsymbol{a}}} \langle T, C \rangle + \sum_{j=1}^n \Omega_{b_j}^{**}(\boldsymbol{t}_j),
\end{aligned}
$$

where we used that strong duality holds, since the conjugate is always convex, even if $\Omega$ is not.

Likewise, for the dual, we have

$$
\begin{aligned}
D_\Omega(\boldsymbol{a}, \boldsymbol{b}, C) &= \max_{\boldsymbol{\alpha} \in \mathbb{R}^m, \boldsymbol{\beta} \in \mathbb{R}^n} \langle \boldsymbol{\alpha}, \boldsymbol{a} \rangle + \langle \boldsymbol{\beta}, \boldsymbol{b} \rangle - \sum_{j=1}^n \Omega_+^*(\boldsymbol{\alpha} + \beta_j \mathbf{1}_m - \boldsymbol{c}_j) \\
&= \max_{\substack{\boldsymbol{\alpha} \in \mathbb{R}^m, \boldsymbol{\beta} \in \mathbb{R}^n \\ \boldsymbol{\mu}_j : \, \boldsymbol{\mu}_j = \boldsymbol{\alpha} + \beta_j \mathbf{1}_m - \boldsymbol{c}_j}} \langle \boldsymbol{\alpha}, \boldsymbol{a} \rangle + \langle \boldsymbol{\beta}, \boldsymbol{b} \rangle - \sum_{j=1}^n \Omega_+^*(\boldsymbol{\mu}_j) \\
&= \max_{\substack{\boldsymbol{\alpha} \in \mathbb{R}^m, \boldsymbol{\beta} \in \mathbb{R}^n \\ \boldsymbol{\mu}_j \in \mathbb{R}^m}} \min_{T \in \mathbb{R}^{m \times n}} \langle \boldsymbol{\alpha}, \boldsymbol{a} \rangle + \langle \boldsymbol{\beta}, \boldsymbol{b} \rangle - \sum_{j=1}^n \Omega_+^*(\boldsymbol{\mu}_j) + \sum_{j=1}^n \langle \boldsymbol{t}_j, \boldsymbol{\mu}_j - \boldsymbol{\alpha} - \beta_j \mathbf{1}_m + \boldsymbol{c}_j \rangle \\
&= \min_{T \in \mathbb{R}^{m \times n}} \langle T, C \rangle + \max_{\boldsymbol{\alpha} \in \mathbb{R}^m} \langle \boldsymbol{\alpha}, T \mathbf{1}_n - \boldsymbol{a} \rangle + \max_{\boldsymbol{\beta} \in \mathbb{R}^n} \langle \boldsymbol{\beta}, T^\top \mathbf{1}_m - \boldsymbol{b} \rangle + \sum_{j=1}^n \max_{\boldsymbol{\mu}_j \in \mathbb{R}^m} \langle \boldsymbol{t}_j, \boldsymbol{\mu}_j \rangle - \Omega_+^*(\boldsymbol{\mu}_j) \\
&= \min_{\substack{T \in \mathbb{R}^{m \times n} \\ T \mathbf{1}_n = \boldsymbol{a} \\ T^\top \mathbf{1}_m = \boldsymbol{b}}} \langle T, C \rangle + \sum_{j=1}^n \Omega_+^{**}(\boldsymbol{t}_j).
\end{aligned}
$$

### B.4 CLOSED-FORM EXPRESSIONS (TABLE 1)

The expressions for the unregularized, negentropy and quadratic cases are provided in (Blondel et al., 2018, Table 1). We therefore focus on the top-$k$ case.

Plugging (11) back into $\langle \boldsymbol{s}, \boldsymbol{t} \rangle - \frac{1}{2}\|\boldsymbol{t}\|_2^2$, we obtain

$$
\Omega_+^*(\boldsymbol{s}) = \sum_{i=1}^k [s_{[i]}]_+ s_{[i]} - \frac{1}{2}[s_{[i]}]_+^2 = \sum_{i=1}^k [s_{[i]}]_+^2 - \frac{1}{2}[s_{[i]}]_+^2 = \frac{1}{2} \sum_{i=1}^k [s_{[i]}]_+^2.
$$

Plugging (12) back into $\langle s, t \rangle - \frac{1}{2}\|t\|_2^2$, we obtain

$$
\begin{aligned}
\Omega_{b_j}^*(s) &= \sum_{i=1}^{k}[s_{[i]} - \tau]_+ s_{[i]} - \frac{1}{2}\sum_{i=1}^{k}[s_{[i]} - \tau]_+^2 \\
&= \sum_{i=1}^{k}\mathbb{1}_{s_{[i]} \geq \tau}(s_{[i]} - \tau)s_{[i]} - \frac{1}{2}\sum_{i=1}^{k}\mathbb{1}_{s_{[i]} \geq \tau}(s_{[i]} - \tau)^2 \\
&= \frac{1}{2}\sum_{i=1}^{k}\mathbb{1}_{s_{[i]} \geq \tau}(s_{[i]}^2 - \tau^2).
\end{aligned}
$$

## B.5 Useful lemmas

**Lemma 1.** *Conjugate of the squared $k$-support norm*

*Let us define the squared $k$-support norm for all $t \in \mathbb{R}^m$ by*

$$
\Psi(t) := \frac{1}{2}\min_{\lambda \in \mathbb{R}^m}\sum_{i=1}^{m}\frac{t_i^2}{\lambda_i} \quad s.t. \quad \langle \lambda, \mathbf{1} \rangle = k, 0 < \lambda_i \leq 1, \forall i \in [m].
$$

*Its conjugate for all $s \in \mathbb{R}^m$ is the squared $k$-support dual norm:*

$$
\Psi^*(s) = \frac{1}{2}\sum_{i=1}^{k}|s|_{[i]}^2.
$$

*Proof.* This result was proved in previous works (Argyriou et al., 2012; McDonald et al., 2016). We include here an alternative proof for completeness.

Using Lapin et al. (2015, Lemma 1), we have for all $a \in \mathbb{R}^m$

$$
\sum_{i=1}^{k}a_{[i]} = \min_{v \in \mathbb{R}} kv + \sum_{i=1}^{m}[a_i - v]_+.
$$

We therefore obtain the variational formulation

$$
\psi(s) := \sum_{i=1}^{k}|s|_{[i]}^2 = \min_{v \in \mathbb{R}} kv + \sum_{i=1}^{m}[|s_i|^2 - v]_+ = \min_{v \in \mathbb{R}} kv + \sum_{i=1}^{m}[s_i^2 - v]_+.
$$

We rewrite the problem in constrained form

$$
\psi(s) = \min_{v \in \mathbb{R}, \xi \in \mathbb{R}_+^m} kv + \langle \xi, \mathbf{1} \rangle \quad s.t. \quad \xi_i \geq s_i^2 - v \quad \forall i \in [m].
$$

We introduce Lagrange multipliers $\lambda \in \mathbb{R}_+^m$, for the inequality constraints but keep the non-negative constraints explicitly

$$
\psi(s) = \min_{v \in \mathbb{R}, \xi \in \mathbb{R}_+^m}\max_{\lambda \in \mathbb{R}_+^m} kv + \langle \xi, \mathbf{1} \rangle + \langle \lambda, s \circ s - v\mathbf{1} - \xi \rangle.
$$

Using strong duality, we have

$$
\begin{aligned}
\psi(s) &= \max_{\lambda \in \mathbb{R}_+^m}\sum_{i=1}^{m}\lambda_i s_i^2 + \min_{v \in \mathbb{R}} v(k - \langle \lambda, \mathbf{1} \rangle) + \min_{\xi \in \mathbb{R}_+^m}\langle \xi, \mathbf{1} - \lambda \rangle \\
&= \max_{\lambda \in \mathbb{R}_+^m}\sum_{i=1}^{m}\lambda_i s_i^2 \quad s.t. \quad \langle \lambda, \mathbf{1} \rangle = k, \lambda_i \leq 1 \, \forall i \in [m].
\end{aligned}
$$

We therefore obtain

$$\psi^*(\boldsymbol{t}) = \max_{\boldsymbol{s} \in \mathbb{R}^m} \langle \boldsymbol{s}, \boldsymbol{t} \rangle - \max_{\boldsymbol{\lambda} \in \mathbb{R}_+^m} \sum_{i=1}^m \lambda_i s_i^2 \quad \text{s.t.} \quad \langle \boldsymbol{\lambda}, \mathbf{1} \rangle = k, \lambda_i \leq 1 \; \forall i \in [m]$$

$$= \min_{\boldsymbol{\lambda} \in \mathbb{R}_+^m} \max_{\boldsymbol{s} \in \mathbb{R}^m} \langle \boldsymbol{s}, \boldsymbol{t} \rangle - \sum_{i=1}^m \lambda_i s_i^2 \quad \text{s.t.} \quad \langle \boldsymbol{\lambda}, \mathbf{1} \rangle = k, \lambda_i \leq 1 \; \forall i \in [m]$$

$$= \frac{1}{4} \min_{\boldsymbol{\lambda} \in \mathbb{R}_+^m} \sum_{i=1}^m \frac{t_i^2}{\lambda_i} \quad \text{s.t.} \quad \langle \boldsymbol{\lambda}, \mathbf{1} \rangle = k, \lambda_i \leq 1 \; \forall i \in [m].$$

Using $\Psi(\boldsymbol{t}) = \frac{1}{2} \psi^*(2\boldsymbol{t})$ gives the desired result. $\qquad \square$

---

**Lemma 2.** *For all $\boldsymbol{s} \in \mathbb{R}^m$ and $b > 0$*

$$\max_{\boldsymbol{t} \in b \triangle^m} \langle \boldsymbol{s}, \boldsymbol{t} \rangle - \frac{1}{2} \|\boldsymbol{t}\|_2^2 = \min_{\theta \in \mathbb{R}} \frac{1}{2} \sum_{i=1}^m [s_i - \theta]_+^2 + \theta b$$

*with optimality condition $\sum_{i=1}^m [s_i - \theta]_+ = b$.*

---

*Proof.*

$$\max_{\boldsymbol{t} \in b \triangle^m} \langle \boldsymbol{s}, \boldsymbol{t} \rangle - \frac{1}{2} \|\boldsymbol{t}\|_2^2 = \max_{\boldsymbol{t} \in \mathbb{R}_+^m} \min_{\theta \in \mathbb{R}} \langle \boldsymbol{s}, \boldsymbol{t} \rangle - \frac{1}{2} \|\boldsymbol{t}\|_2^2 - \theta(\langle \boldsymbol{t}, \mathbf{1} \rangle - b)$$

$$= \max_{\boldsymbol{t} \in \mathbb{R}_+^m} \min_{\theta \in \mathbb{R}} \langle \boldsymbol{s} - \theta \mathbf{1}, \boldsymbol{t} \rangle - \frac{1}{2} \|\boldsymbol{t}\|_2^2 + \theta b$$

$$= \min_{\theta \in \mathbb{R}} \theta b + \max_{\boldsymbol{t} \in \mathbb{R}_+^m} \langle \boldsymbol{s} - \theta \mathbf{1}, \boldsymbol{t} \rangle - \frac{1}{2} \|\boldsymbol{t}\|_2^2$$

$$= \min_{\theta \in \mathbb{R}} \frac{1}{2} \sum_{i=1}^m [s_i - \theta]_+^2 + \theta b.$$

From which we obtain the optimality condition

$$\sum_{i=1}^m [s_i - \theta]_+ = b.$$

$\qquad \square$

### B.6 BICONJUGATES (PROPOSITION 3)

We use $\Omega$ defined in (9).

**Derivation of $\Omega_+^{**}$.**    Recall that

$$\Omega_+^*(\boldsymbol{s}) = \frac{1}{2} \sum_{i=1}^k [s_{[i]}]_+^2.$$

Using Lemma 1, we then have

$$\Omega_+^{**}(\boldsymbol{t}) = \max_{\boldsymbol{s} \in \mathbb{R}^m} \langle \boldsymbol{s}, \boldsymbol{t} \rangle - \Omega_+^*(\boldsymbol{s})$$

$$= \max_{\boldsymbol{s} \in \mathbb{R}^m} \langle \boldsymbol{s}, \boldsymbol{t} \rangle - \frac{1}{2} \sum_{i=1}^{k} [s_{[i]}]_+^2$$

$$= \max_{\boldsymbol{s} \in \mathbb{R}^m, \boldsymbol{\xi} \in \mathbb{R}^m} \langle \boldsymbol{s}, \boldsymbol{t} \rangle - \Psi^*(\boldsymbol{\xi}) \quad \text{s.t.} \quad \boldsymbol{\xi} \geq \boldsymbol{s}$$

$$= \max_{\boldsymbol{s} \in \mathbb{R}^m, \boldsymbol{\xi} \in \mathbb{R}^m} \min_{\boldsymbol{\mu} \in \mathbb{R}_+^m} \langle \boldsymbol{s}, \boldsymbol{t} \rangle - \Psi^*(\boldsymbol{\xi}) - \langle \boldsymbol{\mu}, \boldsymbol{s} - \boldsymbol{\xi} \rangle$$

$$= \min_{\boldsymbol{\mu} \in \mathbb{R}_+^m} \max_{\boldsymbol{s} \in \mathbb{R}^m} \langle \boldsymbol{s}, \boldsymbol{t} - \boldsymbol{\mu} \rangle + \max_{\boldsymbol{\xi} \in \mathbb{R}^m} \langle \boldsymbol{\mu}, \boldsymbol{\xi} \rangle - \Psi^*(\boldsymbol{\xi})$$

$$= \begin{cases} \Psi(\boldsymbol{t}) & \text{if } \boldsymbol{t} \in \mathbb{R}_+^m \\ \infty, & \text{otherwise} \end{cases}.$$

Note that $\boldsymbol{\xi}$ is not constrained to be non-negative because it is squared in the objective.

**Derivation of $\Omega_b^{**}$.** Recall that

$$\Omega_b^*(\boldsymbol{s}) := \max_{\boldsymbol{t} \in b \triangle^m \cap \mathcal{B}_k} \langle \boldsymbol{s}, \boldsymbol{t} \rangle - \frac{1}{2} \|\boldsymbol{t}\|_2^2.$$

Using Lemma 1, Lemma 2 and (12), we obtain

$$\Omega_b^*(\boldsymbol{s}) = \min_{\tau \in \mathbb{R}} \frac{1}{2} \sum_{i=1}^{k} [s_{[i]} - \tau]_+^2 + \tau b$$

$$= \min_{\tau \in \mathbb{R}, \boldsymbol{\xi} \in \mathbb{R}^m} \Psi^*(\boldsymbol{\xi}) + \tau b \quad \text{s.t.} \quad \boldsymbol{\xi} \geq \boldsymbol{s} - \tau \mathbf{1}$$

$$= \min_{\tau \in \mathbb{R}, \boldsymbol{\xi} \in \mathbb{R}^m} \max_{\boldsymbol{\mu} \in \mathbb{R}_+^m} \Psi^*(\boldsymbol{\xi}) + \tau b + \langle \boldsymbol{\mu}, \boldsymbol{s} - \tau \mathbf{1} - \boldsymbol{\xi} \rangle.$$

We then have

$$\Omega_b^{**}(\boldsymbol{t}) = \max_{\boldsymbol{s} \in \mathbb{R}^m} \langle \boldsymbol{s}, \boldsymbol{t} \rangle - \Omega_b^*(\boldsymbol{s})$$

$$= \min_{\boldsymbol{\mu} \in \mathbb{R}_+^m} \max_{\tau \in \mathbb{R}} \tau(\langle \boldsymbol{\mu}, \mathbf{1} \rangle - b) + \max_{\boldsymbol{s} \in \mathbb{R}^m} + \langle \boldsymbol{s}, \boldsymbol{t} - \boldsymbol{\mu} \rangle + \max_{\boldsymbol{\xi} \in \mathbb{R}^m} \langle \boldsymbol{\xi}, \boldsymbol{\mu} \rangle - \Psi^*(\boldsymbol{\xi})$$

$$= \begin{cases} \Psi(\boldsymbol{t}) & \text{if } \boldsymbol{t} \in b \triangle^m \\ \infty, & \text{otherwise} \end{cases}.$$

**Proof of proposition.** We recall that $\Psi$ is defined in (13). From Proposition 2, we have $D_\Omega(\boldsymbol{a}, \boldsymbol{b}, C) = S_\Omega(\boldsymbol{a}, \boldsymbol{b}, C) = P_\Psi(\boldsymbol{a}, \boldsymbol{b}, C)$. From Proposition 1 (weak duality), we have $P_\Psi(\boldsymbol{a}, \boldsymbol{b}, C) \leq P_\Omega(\boldsymbol{a}, \boldsymbol{b}, C)$.

Assuming no ties in $\boldsymbol{\alpha}^\star + \beta_j^\star \mathbf{1}_m - \boldsymbol{c}_j$ or in $\boldsymbol{\alpha}^\star - \boldsymbol{c}_j$ for all $j \in [n]$, we know that $\boldsymbol{t}_j^\star \in \mathcal{B}_k$ for all $j \in [n]$. Furthermore, from (13), we have for all $\boldsymbol{t}_j \in \mathcal{B}_k$ that $\Omega(\boldsymbol{t}_j) = \Psi(\boldsymbol{t}_j) = \frac{1}{2} \|\boldsymbol{t}_j\|_2^2$. Therefore, without any ties, we have $P_\Psi(\boldsymbol{a}, \boldsymbol{b}, C) = P_\Omega(\boldsymbol{a}, \boldsymbol{b}, C)$.

### B.7 LIMIT CASES (PROPOSITION 4)

In the limit case $k = 1$ with $\Omega$ defined in (9), we have

$$\Omega_b^*(\boldsymbol{s}) = \max_{\boldsymbol{t} \in b \triangle^m \cap \mathcal{B}_1} \langle \boldsymbol{s}, \boldsymbol{t} \rangle - \frac{\gamma}{2} \|\boldsymbol{t}\|_2^2$$

$$= \max_{\boldsymbol{t} \in b\{\boldsymbol{e}_1, \dots, \boldsymbol{e}_m\}} \langle \boldsymbol{s}, \boldsymbol{t} \rangle - \frac{\gamma}{2} \|\boldsymbol{t}\|_2^2$$

$$= b \max_{\boldsymbol{t} \in \{\boldsymbol{e}_1, \dots, \boldsymbol{e}_m\}} \langle \boldsymbol{s}, \boldsymbol{t} \rangle - \frac{\gamma}{2} b^2$$

$$= b \max_{i \in [m]} s_i - \frac{\gamma}{2} b^2.$$

We therefore get

$$
\begin{aligned}
S_\Omega(\boldsymbol{a}, \boldsymbol{b}, C) &= \max_{\boldsymbol{\alpha} \in \mathbb{R}^m} \langle \boldsymbol{a}, \boldsymbol{\alpha} \rangle - \sum_{j=1}^{n} \Omega_{b_j}^*(\boldsymbol{\alpha} - \boldsymbol{c}_j) \\
&= \max_{\boldsymbol{\alpha} \in \mathbb{R}^m} \langle \boldsymbol{a}, \boldsymbol{\alpha} \rangle - \sum_{j=1}^{n} b_j \max_{i \in [m]} \alpha_i - c_{i,j} + \frac{\gamma}{2} \sum_{j=1}^{n} b_j^2 \\
&= S_0(\boldsymbol{a}, \boldsymbol{b}, C) + \frac{\gamma}{2} \sum_{j=1}^{n} b_j^2.
\end{aligned}
$$

Likewise, we have

$$
\begin{aligned}
\Omega_+^*(\boldsymbol{s}) &= \max_{\boldsymbol{t} \in \mathbb{R}_+^m \cap \mathcal{B}_1} \langle \boldsymbol{s}, \boldsymbol{t} \rangle - \frac{\gamma}{2} \|\boldsymbol{t}\|_2^2 \\
&= \max_{i \in [m]} \max_{t \in \mathbb{R}_+} s_i t - \frac{\gamma}{2} t^2 \\
&= \frac{1}{2\gamma} \max_{i \in [m]} [s_i]_+^2.
\end{aligned}
$$

We therefore get

$$
\begin{aligned}
D_\Omega(\boldsymbol{a}, \boldsymbol{b}, C) &= \max_{\boldsymbol{\alpha} \in \mathbb{R}^m, \boldsymbol{\beta} \in \mathbb{R}^n} \langle \boldsymbol{\alpha}, \boldsymbol{a} \rangle + \langle \boldsymbol{\beta}, \boldsymbol{b} \rangle - \sum_{j=1}^{n} \Omega_+^*(\boldsymbol{\alpha} + \beta_j \mathbf{1}_m - \boldsymbol{c}_j) \\
&= \max_{\boldsymbol{\alpha} \in \mathbb{R}^m, \boldsymbol{\beta} \in \mathbb{R}^n} \langle \boldsymbol{\alpha}, \boldsymbol{a} \rangle + \langle \boldsymbol{\beta}, \boldsymbol{b} \rangle - \frac{1}{2\gamma} \sum_{j=1}^{n} \max_{i \in [m]} [\alpha_i + \beta_j - c_{i,j}]_+^2.
\end{aligned}
$$

From Proposition 3, we have $D_\Omega = S_\Omega = S_0 + \frac{\gamma}{2} \|b\|_2^2 = P_0 + \frac{\gamma}{2} \|b\|_2^2$.

