# OpenReview forum: "Sparsity-Constrained Optimal Transport"
_ICLR.cc/2023/Conference — ICLR 2023 notable top 25%_

### Official Review · Reviewer_yCJL · 2022-10-23

**Confidence:** 4
**Correctness:** 4
**Technical Novelty And Significance:** 4
**Empirical Novelty And Significance:** 4
**Recommendation:** 10

**Clarity, Quality, Novelty And Reproducibility:**

The idea of adding direct control to the sparsity by adding cardinality constraints to the columns of transportation plans is very novel and effective, this greatly improves the sparsity and the accuracy of optimal transportation plans.

The work is very carefully prepared, very well written. All the concepts, terminologies, theoretic formulations and main claims are carefully formulated, rigorously proved, and vividly illustrated and thoroughly discussed. Although the work involves advanced background knowledge in optimal transportation theory, convex analysis and optimization, the manuscript is relatively easy to follow and pleasant to study.

The manuscript has very good quality. The mathematical models are elegant, the theoretical methods are powerful, the proofs are rigorous and the experimental results are convincing.

The work should be easy to reproduce, all the algorithmic details are explained clearly.

**Details Of Ethics Concerns:**

This work is a fundamental research, focuses mainly on theoretic exploration.

**Strength And Weaknesses:**

The work has many strengths:
1. The problem to tackle is fundamental: control the sparsity of optimal transportation plans, which is crucial for deep learning.
2. The idea of adding direct control to the cardinality of the images of transportation plan is novel and effective.
3. The high level perspective, which unifies the proposed method and the existing ones, is inspiring and promising.
4. The work has solid theoretic foundation, all the main claims are formulated as theorems and proved rigorously.
5. The experimental results are thorough and convincing.

The weakness of the work is that it requires advanced background knowledge,  currently, detailed discussions in the supplementary materials are very helpful, it will be more helpful to give more explanations to the key concepts and the main theorems.



**Summary Of The Paper:**

Entropy-regularized optimal transport is very popular today, but it leads to fully-dense transport plans. To address this problem, this work proposes a new approach for OT with explicit cardinality constraints on the transportation plan. Despite the non-convexity of the cardinality constraints, the work shows that the corresponding dual or semi-dual problems are tractable and can be solved with first-order gradient methods. This method can be treated as a middle ground between unregularized OT (case k=1) and quadratically-regularized OT( k is big enough). The smoothness of the objectives increases as k increases, giving rise to a trade-off between convergence speed and sparsity of the optimal plan.

The contributions are
1. presents a dual and semi-dual framework for OT with general nonconvex regularization. The framework is applied to obtain a tractable lower bound to solve an OT formulation with cardinality constraints on the transportation plan.
2. shows this framework is formally equivalent to using squared k-support norm regularization in the primal, this model is the middle point between the unregularized OT and the quadratically-regularized OT.
3. The (semi) dual objectives were shown to be increasingly smooth as k increases, enabling the use of gradient-based algorithms such as LBFGS or ADAM.
4. the framework is demonstrated on a variety of tasks, which shows a direct control of sparsity improves the accuracy, lead to more interpretable transportation plans and the integer-valued hyperparameter k may be easier to tune.


**Summary Of The Review:**

This work presents a dual and semi-dual framework for OT with general nonconvex regularization. The framework is applied to obtain a tractable lower bound to solve an OT formulation with cardinality constraints on the transportation plan. The work also shows this framework is formally equivalent to using squared k-support norm regularization in the primal, this model is the middle point between the unregularized OT and the quadratically-regularized OT. The (semi) dual objectives were shown to be increasingly smooth as k increases, enabling the use of gradient-based algorithms such as LBFGS or ADAM. The framework is demonstrated on a variety of tasks, which shows a direct control of sparsity improves the accuracy, lead to more interpretable transportation plans and the integer-valued hyperparameter k may be easier to tune.

The sparsity problem is fundamental in deep learning, this work gives a nice solution to the problem. The work has solid theoretic foundation, and high practical value. The mathematical formulation and proofs are rigorous and elegant. The description and analysis are clear and easy to follow. The experimental results are encouraging. Especially, this work unifies the existing methods, and present a coherent theoretic framework, this global view is very valuable.

---

> ### Author Response · Authors · 2022-11-08
> **Thank you**
>
> Thank you for the extremely positive feedback. We are very happy that you enjoyed the paper.

---

### Official Review · Reviewer_1nGA · 2022-10-25

**Confidence:** 4
**Correctness:** 4
**Technical Novelty And Significance:** 3
**Empirical Novelty And Significance:** 3
**Recommendation:** 8

**Clarity, Quality, Novelty And Reproducibility:**

- The paper is well-written and easy to follow. All the proofs are complete.
- Code is not available, but I think reproducibility could be guaranteed since the authors detail the setting of each experiment.


**Strength And Weaknesses:**

### Strength ###
- To the best of my knowledge, working with nonconvex regularization on OT problems is not well-known. This paper investigates the $k$-cardinality constraint, that forces each column to get at most $k$ non-zeros elements, avoids the spreading of masses in the OT solution. I really appreciate the finite analytic results about the relations between the primal, dual, and semi-dual forms of the objective functions.
- The approach is strongly supported by theoretical findings.
- As the OT community is a big fan of acronyms, so I can suggest the following acronym for this paper **Top-$k$-OT**.
- Extensive numerical experiments are conducted on sparse mixture experts with good performances and on supply-demand transportation spherical data.

I have a single question: what is the computational complexity of **Top-$k$ OT**?  One can complete Table 2 by adding time for the proposed algorithm.

**Summary Of The Paper:**

Optimal transport (OT) distances (a.k.a. Wasserstein's distances) are a powerful computational tool to compare probability distributions and they play an increasingly preponderant role in machine learning, statistics, and computer vision. Since computing OT distances involves a linear program, which takes super-cubic time in the data size, its large-scale applications have mainly been hampered by its high computational cost. A classical and popular approach to tackling this problem consists in adding an entropic regularization that can be solved using Sinkhorn iterations. Comparing to the (unregularized) OT that produces a sparse transport plan, the entropic penalty blurs the OT solution, namely, it entails a dense transport plan being undesirable for interpretability. Towards realizing a sparse solution, Blondel et al. 2018 (ICML) proposed to substitute the entropy penalty with the squared-$2$norm.

In this spirit of obtaining a sparse OT, this paper proposes a novel approach by working on a nonconvex penalty with control of the sparsity. In other words, the authors define the $k$-cardinality constraint penalty that controls the cardinality of non-zeros elements in each column of the transport plan. The idea is quite "nice" and seems interesting for many applications involving OT metrics.

**Summary Of The Review:**

Overall, I like the approach and I think it deserves to be published at ICLR conference since it gives advances to the OT community.

---

> ### Author Response · Authors · 2022-11-08
> **Thank you**
>
> Thank you for the very positive feedback and valuable suggestions. We are very happy that you enjoyed the paper.
>
> > I have a single question: what is the computational complexity of Top-k OT?
>
> For the dual, for which the conjugate only involves non-negativity constraints, the complexity per LBFGS or ADAM iteration is $O(mn)$, whatever the regularization used.
>
> For the semi-dual, for which the conjugate involves simplex constraints, the complexity per LBFGS or ADAM iteration is as follows:
>
> - Sparsity-constrained: $O(mn \\log k)$, due to the k-sparse projection on the simplex
> - Quadratic regularization: $O(mn \\log m)$, due to the projection on the simplex
> - Entropic regularization: $O(mn)$, due to the softmax
>
> > Code is not available, but I think reproducibility could be guaranteed since the authors detail the setting of each experiment.
>
> We plan to open-source our JAX code upon publication of this paper.
>
> > As the OT community is a big fan of acronyms, so I can suggest the following acronym for this paper Top-k OT
>
> Thank you for the suggestion. We are a bit concerned that this title could be misinterpreted. For example, there is a paper titled “Differentiable Top-k Operator with Optimal Transport” (https://arxiv.org/abs/2002.06504). Top-k OT could also be interpreted as finding the top-k solutions of the OT problem. To avoid these potential confusions, we prefer to stick to “Sparsity-constrained OT”.

---

> > ### Comment · Reviewer_1nGA · 2022-11-19
> > **Thank you**
> >
> > I thank the authors for their response to my question.

---

### Official Review · Reviewer_hZUP · 2022-10-29

**Confidence:** 4
**Correctness:** 4
**Technical Novelty And Significance:** 2
**Empirical Novelty And Significance:** 2
**Recommendation:** 8

**Clarity, Quality, Novelty And Reproducibility:**

Very clear, writing is high quality, no concerns about reproducibility but concerns in novelty as explained in the weaknesses section.

**Strength And Weaknesses:**

Strengths:

-- Largely well written and the relevant works are well-cited.
-- The duality connection is well-explained, albeit is standard for solving such problems. Similarly, the section explaining the development of the dual specific to the sparsity constrained problem is also well-explained.

Weaknesses:

-- I have concerns for novelty. Given enormous progress already made with iterative hard thresholding and its variants, I am not sure if they optimal transport setup adds enough novelty above that. I suggest the authors look at works of Anastasios Kyrillidis (one of whose papers the authors have cited) to look at iterative hard thresholding methods that are directly employed on the primal. Given those works, I am unfortunately not excited about this paper.

-- Similarly the section on double conjugate mentioning what problem is solved when solving the dual and the relationship with k-support norms is known.

-- There are limited empirical evaluations, especially considering the relevance to this conference.



**Summary Of The Paper:**

The paper proposes an optimization algorithm for sparse optimal transport. The optimal transport is a linear program with additional regularization on the columns on the transportation plan matrix. In this work, the authors add further constraints to ensure each column has no more than k non-zeroes. They then propose using gradient based optimizers to directly solve the non-convex problem.

**Summary Of The Review:**

After the rebuttal and reading other reviews, I am updating my score. I seem to have misunderstood the paper's contributions earlier.

Solving non-convex problems using convex duals is not new, the paper explains the setup for sparse constrained optimization in context of optimal transport. I have concerns for novelty, limited empirical evaluation and the relevance to the iclr community.

---

> ### Author Response · Authors · 2022-11-08
> **Clarifications**
>
> Thank you for your review. As we point out below, primal algorithms would be very costly to apply due to the transportation plan constraint $T \in \mathcal{U}(a, b)$. Regarding the link with the k-support norm, what makes our Proposition 3 new and non-trivial are the non-negativity or simplex constraints.
>
> > The duality connection is well-explained, albeit is standard for solving such problems.
> > Solving non-convex problems using convex duals is not new, the paper explains the setup for sparse constrained optimization in context of optimal transport.
>
> Using weak duality for approximately solving a nonconvex problem is a known technique but it requires being able to compute the conjugate (equation 10 in our case). Therefore, it can only be applied when the nonconvex problem has a particular structure. Since this technique is not applicable to any nonconvex problem, we therefore argue that our application of weak duality to our particular problem is valuable.
>
> > I have concerns for novelty. Given enormous progress already made with iterative hard thresholding and its variants, I am not sure if they optimal transport setup adds enough novelty above that. I suggest the authors look at works of Anastasios Kyrillidis (one of whose papers the authors have cited) to look at iterative hard thresholding methods that are directly employed on the primal. Given those works, I am unfortunately not excited about this paper.
>
> We emphasize that primal algorithms such as iterative hard thresholding would require a projection step due to the constraint $T \in \mathcal{U}(a, b)$. This projection step does not enjoy a closed form and can be recast as an $\ell_2$-regularized OT problem itself. This means that the projection step is as difficult as an $\ell_2$-regularized OT problem and that we would need to solve it at every iteration of iterative hard thresholding. Clearly, this would be horribly slow. In fact, using regularization and switching to the dual or semi-dual is precisely to avoid having to deal with the $T \in \mathcal{U}(a, b)$ constraint.
>
> > Similarly the section on double conjugate mentioning what problem is solved when solving the dual and the relationship with k-support norms is known.
>
> It is difficult to check the reviewer’s claim, since a reference was not provided.
>
> To our knowledge, what is a known result is that the k-support unit-ball is the convex hull of $\\{x \in \mathbb{R}^n \colon \\|x\\|_0 \le k, \\|x\\|_2 \le 1\\}$. See, e.g., Argyriou et al. 2012 or McDonald et al., 2016 cited in the paper. See also “The k-Support Norm and Convex Envelopes of Cardinality and Rank” by Eriksson et al.
>
> Our Proposition 3 is a different result. Formally, using the same notation as in the paper, it shows that if we set
>
> $f =  \\delta_{\\mathcal{C}} + \\delta_{\\mathcal{B}_k} + \\frac{1}{2} \\|\cdot\\|^2_2$,
>
> where $\mathcal{C}$ is either the non-negative orthant or the simplex, then
>
> $f^{**} = \Psi + \delta_{\mathcal{C}}$,
>
> where $\Psi$ is the squared k-support norm. Clearly, this result does not hold for any convex set $\mathcal{C}$. We argue that our Proposition 3 does not trivially follow from the known result of Argyriou et al (e.g. we cannot use simple tools such as infimal convolutions).
> If the reviewer thinks that this is a known result (with the non-negative orthant or simplex constraints), we kindly ask for an actual reference.
>
> > I have concerns for limited empirical evaluation
>
> We would appreciate it if the reviewer could be more specific and point out which empirical evaluation is perceived as limited and in what sense. Otherwise, it can be challenging to address the reviewer’s concern.
> To recap our current experimental results, we conducted MoE experiments on large-scale vision tasks and obtained impressive few-shot image recognition results on ImageNet. We also conducted numerous additional experiments including supply-demand transportation, color transfer, and balanced image clustering (see Appendix A).  Our experiments were supported by all other reviewers – reviewer *aL1u*: “The paper provides pretty strong application motivation of the sparsity-constrained optimal transport, e.g. the sparse mixture of experts”, reviewer *xd11*: “The extensive experiments help show the performance of the proposed method”), reviewer *1nGA*: “Extensive numerical experiments are conducted on sparse mixture experts with good performances and on supply-demand transportation spherical data”, and reviewer *yCJL*: “The experimental results are thorough and convincing”.
>
> > and the relevance to the iclr community
>
> We respectfully disagree. The theoretical contribution of our work is a new OT formulation, which is relevant to the optimization and OT community at ICLR. The main application contribution of our work is sparse mixture of experts (MoE), which is highly relevant to the machine learning and computer vision community at ICLR.

---

> > ### Comment · Reviewer_hZUP · 2022-11-16
> > **Thanks**
> >
> > Thank you for clarifications. I have revised my score, as I seem to have not grasped the contributions before.

---

> ### Author Response · Authors · 2022-11-15
> **Could you please check our response?**
>
> Dear Reviewer,
>
> Since only two days are left in the discussion period, we would appreciate it if you could check and reply to our response to your comments soon. We believe we have addressed all your major comments:
>
> - Primal algorithms such as iterative hard thresholding are not applicable due to the transportation plan constraint $T \in \mathcal{U}(a, b)$. Switching to the dual is crucial to avoid having to deal with this constraint.
> - Proposition 3 linking the biconjugate with the squared $k$-support norm is novel and non-trivial due to the non-negativity or simplex constraints.
> - Our experiments include large-scale sparse mixture of expert on vision tasks, as well as color transfer, balanced clustering and supply-demand allocation.
> - Optimal transport and sparse mixture of experts are highly relevant to the ICLR community.
>
> If our response adequately addresses your concerns, please consider raising the score of our submission.
>
> Thank you very much for your time.

---

### Official Review · Reviewer_xd11 · 2022-10-29

**Confidence:** 4
**Correctness:** 4
**Technical Novelty And Significance:** 3
**Empirical Novelty And Significance:** 3
**Recommendation:** 6

**Clarity, Quality, Novelty And Reproducibility:**

Generally, the paper is clear and easy to follow. Different sections are well organized and help support each other. The proposed method is novel and there is no reproducibility problem.

**Strength And Weaknesses:**

Strength
- It is novel to add the k-sparsity constraint to the regularized OT problem and use the k-sparse projection to make the solution tractable.
- To reconstruct the OT plan, the use of bi-conjugate tech is interesting.
- The extensive experiments help show the performance of the proposed method.

Weakness
- Is there any guarantee for the sparsity of the reconstructed OT plan through bi-conjugate?
- Since the authors are trying to solve an OT problem, it is necessary to investigate the influence of $\gamma$ in equation (8). I am extremely curious about the convergence/performance with a small $\gamma$. Ablation study is needed here.
- Besides sparsity, the accuracy of the propose method should also be reported. With small scale problems like $784\times 784$, the authors can use linear programming to obtain the groundtruth and make the comparison.
- It is important to report the scale of the problems in the experiments. For small scale problems, linear programming may be more efficient.
- Lack of important reference paper "An et al. Efficient Optimal Transport Algorithm by Accelerated Gradient descent, AAAI 2022". The authors may also try the accelerated gradient decent method to see if it helps improve the performance.

**Summary Of The Paper:**

In this paper, the authors propose to add a non-convex sparsity constraint to the $l^2$ regularized discrete OT problem. In such a way, they can control the sparsity of the OT plan. Though the sparsity constraint makes the original problem nonconvex, it is still tractable through the dual and semi-dual problem. With the k-sparse projection, the proposed problem can be solved by L-BFGS or ADAM. Furthermore, to obtain the approximate OT plan, they turn to the convex bi-conjugate problem. Experiments show that the proposed method successfully induce a controlled sparse solution.

**Summary Of The Review:**

The paper propose a novel method to solve the discrete OT problem with controlled sparsity. Experiments show that the proposed method work well. To make the results more convincible

---

> ### Author Response · Authors · 2022-11-08
> **Answers to comments**
>
> Thank you for your review and comments.
>
> > Is there any guarantee for the sparsity of the reconstructed OT plan through bi-conjugate?
>
> By equations 10 and 11, which are used to recover the OT plan (see also equations 6 and 7), the maximum sparsity $k$ is guaranteed to be respected, as long as $k$ is not so small that the marginal constraints cannot be respected.
>
> > Since the authors are trying to solve an OT problem, it is necessary to investigate the influence of γ in equation (8). I am extremely curious about the convergence/performance with a small γ. Ablation study is needed here.
>
> In general, the reviewer is right that the larger $\gamma$, the faster LBFGS or ADAM should converge. However, note that if we multiply $\gamma$ by some constant, then the OT plan is not changed if we divide the cost matrix $C$ by the same constant. In our sparse MoE experiments, the cost matrix is generated by a learned bilinear model, so that constant can be absorbed without loss of generality.
>
> >Besides sparsity, the accuracy of the proposed method should also be reported. With small scale problems like 784×784, the authors can use linear programming to obtain the groundtruth and make the comparison.
>
> Empirically checking if our proposed approach is close to the unregularized solution would be interesting but it is not our goal in this paper, since our goal is mainly to find a good router mechanism for sparse MoEs, which is yet easy to implement on GPU and TPU, unlike linear programming solvers.
>
> > It is important to report the scale of the problems in the experiments. For small scale problems, linear programming may be more efficient.
>
> While this is true in principle, solvers for the linear programming formulation of OT are usually not GPU or TPU friendly. This is typically one of the motivations for introducing regularization.
>
> > Lack of important reference paper "An et al. Efficient Optimal Transport Algorithm by Accelerated Gradient descent, AAAI 2022". The authors may also try the accelerated gradient decent method to see if it helps improve the performance.
>
> We added the reference, thank you.

---

> > ### Comment · Reviewer_xd11 · 2022-11-16
> > **Thanks and one more question**
> >
> > I thank the authors for the detailed reply, and most of them are convincible. To investigate the influence of $\gamma$, we need to fix $C$ and change $\gamma$, not to divide them with the same factor. As a key parameter, some ablation study of $\gamma$ is necessary and can help readers better understand the problem.

---

> > > ### Author Response · Authors · 2022-11-16
> > > **Clarification**
> > >
> > > We just meant that when $C$ is generated by a linear model (which is the case in our sparse MoE experiment), $\gamma$ can be absorbed into the model. This is because the argmin of $\langle T, C \rangle - \gamma \Omega(T)$ is the same as the argmin of $\langle T, C / \gamma \rangle - \Omega(T)$. Hence if $C$ is generated by a linear model, $\gamma$ can be absorbed in that model. When $C$ is fixed, we agree that investigating the effect of $\gamma$ on convergence speed would make sense. We expect it to have the same effect as for entropic regularization or quadratic regularization.

---

### Official Review · Reviewer_aL1u · 2022-11-05

**Confidence:** 4
**Correctness:** 3
**Technical Novelty And Significance:** 3
**Empirical Novelty And Significance:** 3
**Recommendation:** 6

**Clarity, Quality, Novelty And Reproducibility:**

Clarity: The paper is overall well-written, with clear structure and plenty of discussion. Some unclear points are listed in weaknesses.
Quality: The paper is of decent quality, where the writing is consistent and the presentation is not hard to follow.
Novelty: The paper contains some novel combination of interesting ideas, and provides notable results that offers new insight to the sparse OT problem.
Reproducibility: the experiments can be reproduced.

**Strength And Weaknesses:**

The paper has the following strengths:
1. The paper answers, in some sense, the question of how to make optimal transport plan more sparse, especially with proposition 3 where the paper introduces the k-support norm, which is of general interest and is a more principled way to enforce sparsity.
2. The most notable result in the reviewer's opinion, is that it reveals the relations between the k-support norm regularization, and the sparsity of the OT plan. Though the strong duality does not hold, the sparsity constraint is still nicely carried from the original primal problem into the dual solution.
3. The paper provides pretty strong application motivation of the sparsity-constrained optimal transport, e.g. the sparse mixture of experts.

Some weaknesses:
1. Though the paper claim it proposed a 'constrained' problem, it does not solve the primal constrained problem exactly. In fact, the paper finds feasible solution(s) to the original problem through minimax principle, but no approximate optimality is provided. It would be more natural if some tightness result between primal and dual formulations can be provided, as for the current paper the 'constraint' point of view is more of a heuristic formulation rather than a problem solved in the paper. (Maybe considering the tightness between $\ell_2$ and $\Psi$ can lead to it.)
2. The paper claimed that the proposed formulation has increased smoothness when the constraint is looser, but the reviewer didn't find a concrete characterization of the increase of smoothness except for a figure illustrating this phenomenon. This point would be more valid and clear if some concrete measurement of smoothness can be added, and even a formal proof of such increase of smoothness.

Minor comment: the paper claimed in [section 3][computation] that the entropic formulation is not smooth, which is unclear to the reviewer, since it is well known that the dual of entropic OT is sum of exponentiation of dual potentials which is very smooth, and the whole quantity is even smooth in marginals, see e.g. https://arxiv.org/pdf/2207.08683.pdf , https://arxiv.org/pdf/2207.07427.pdf .

**Summary Of The Paper:**

The paper studies optimal transport with prescribed sparsity. Specifically, based on the well known quadratic regularized optimal transport, the paper further constrains the cardinality of non-zeros entries of each column of the optimal plan. The problem of optimal transport with constrained sparsity is well motivated from expert assignment problem, and based on weak duality of the proposed optimization problem, efficient algorithm is proposed, with experiments provided.

**Summary Of The Review:**

The paper is overall well written and organized. Notable results are proven, with novel regularization that leads to a desired level of sparsity. Some weaknesses are listed for possible improvement. In summary The paper is a solid contribution to the sparse optimal transport problem.

---

> ### Author Response · Authors · 2022-11-08
> **Answers to comments**
>
> Thank you for taking the time to review our paper and for the valuable comments.
>
> > The most notable result in the reviewer's opinion, is that it reveals the relations between the k-support norm regularization, and the sparsity of the OT plan. Though the strong duality does not hold, the sparsity constraint is still nicely carried from the original primal problem into the dual solution.
>
> Indeed, strong duality does not hold when using the nonconvex sparsity constraint but we would like to clarify that strong duality does hold when using the squared k-support norm $\Psi$ as regularization.
>
> > Though the paper claim it proposed a 'constrained' problem, it does not solve the primal constrained problem exactly. In fact, the paper finds feasible solution(s) to the original problem through minimax principle, but no approximate optimality is provided. It would be more natural if some tightness result between primal and dual formulations can be provided, as for the current paper the 'constraint' point of view is more of a heuristic formulation rather than a problem solved in the paper. (Maybe considering the tightness between ℓ2 and Ψ can lead to it)
>
> We agree that such approximation guarantees would be a great addition, though this seems to be very challenging. However, we already know that the biconjugate is the tightest convex relaxation. Therefore, our approach uses the optimal convex relaxation.
>
> > The paper claimed that the proposed formulation has increased smoothness when the constraint is looser, but the reviewer didn't find a concrete characterization of the increase of smoothness except for a figure illustrating this phenomenon. This point would be more valid and clear if some concrete measurement of smoothness can be added, and even a formal proof of such increase of smoothness.
>
> This is also a very valid comment, though this is again very challenging. We leave this theoretical investigation to future work.
>
> > Minor comment: the paper claimed in [section 3][computation] that the entropic formulation is not smooth, which is unclear to the reviewer, since it is well known that the dual of entropic OT is sum of exponentiation of dual potentials which is very smooth, and the whole quantity is even smooth in marginals, see e.g. https://arxiv.org/pdf/2207.08683.pdf , https://arxiv.org/pdf/2207.07427.pdf
>
> In the above two papers, they use “smooth” in the sense of “infinitely differentiable”.
>
> Throughout our paper, we use “smooth” in the sense of having Lipschitz gradients, which is equivalent to having upper-bounded second derivatives. It is easy to check that the second derivatives of the dual objective of entropy-regularized OT are indeed unbounded, while the second derivatives of the semi-dual objective are bounded.
>
> Alternatively, this can also be seen from the duality between strong convexity and smoothness. Indeed, the Shannon entropy is not strongly convex on the non-negative orthant but it is on the simplex (boundedness of the domain is crucial in the strong convexity proof). This implies that the dual OT problem is not smooth, but that the semi-dual is. With squared $\ell_2$ regularization, we have strong convexity over any convex set. This implies that both the dual and the semi-dual are smooth with that regularization.

---

> > ### Comment · Reviewer_aL1u · 2022-11-16
> > **Comment**
> >
> > Thanks for the reply. The arguments seem valid and I will leave my score unchanged.

---

### Decision · Program_Chairs · 2023-01-20

**Decision:**

Accept: notable-top-25%

**Justification For Why Not Higher Score:**

 The techniques proposed in the paper  should be highlighted especially in circumventing iterative thresholding and resorting to dual and semi dual instead to yield an efficient algorithm to impose sparsity on OT plan. The paper does not meet the bar of an oral since it is missing a complete theoretical study of the tightness of the feasible point returned by the algorithm when solving the dual or semi dual due to non convexity.

**Justification For Why Not Lower Score:**

All reviewers agreed on the novelty and the quality of the work.

**Metareview: Summary, Strengths And Weaknesses:**

The paper proposes a sparsity constraint on the optimal transport plan and resort to weak duality (and semi duals) to find an approximation of the original primal problem. The bi-conjugate maps are used to reconstruct the optimal and  experimentation shows the  benefits of the methods in recovery of sparse transport plans.

The main weakness of the work is the lack of approximation guarantees for the algorithm proposed in the paper , since it is a non-convex optimization. This does not however hinder the value of the work but leaves room for future investigation.

**Note From Pc:**

if the above contains the word "oral" or "spotlight" please see: "oral" presentation means -> notable-top-5% and "spotlight" means -> notable-top-25%. As stated in our emails, we are disassociating presentation type from AC recommendations

**Summary Of Ac-Reviewer Meeting:**

N/A